# STRICTLY CONSTRAINED GENERATIVE MODELING VIA SPLIT AUGMENTED LANGEVIN SAMPLING

**Matthieu Blanke** *
New York University
LEAP NSF STC

**Yongquan Qu**
Columbia University
LEAP NSF STC

**Sara Shamekh**
New York University
LEAP NSF STC

**Pierre Gentine**
Columbia University
LEAP NSF STC

## ABSTRACT

Deep generative models hold great promise for representing complex physical systems, but their deployment is currently limited by the lack of guarantees on the physical plausibility of the generated outputs. Ensuring that known physical constraints are enforced is therefore critical when applying generative models to scientific and engineering problems. We address this limitation by developing a principled framework for sampling from a target distribution while rigorously satisfying mathematical constraints. Leveraging the variational formulation of Langevin dynamics and Lagrangian duality, we propose Constrained Alternated Split Augmented Langevin (CASAL), a novel primal-dual sampling algorithm that enforces constraints progressively through variable splitting. We analyze our algorithm in Wasserstein space and derive explicit mixing time rates. While the method is developed theoretically for Langevin dynamics, we demonstrate its applicability to diffusion models. We apply our method to diffusion-based data assimilation on a complex physical system, where enforcing physical constraints substantially improves both forecast accuracy and the preservation of critical conserved quantities. We also demonstrate the potential of CASAL for challenging non-convex feasibility problems in optimal control.

## 1 INTRODUCTION

Generative deep learning methods have recently emerged as powerful tools to model and sample from complex data distributions, with successful applications in image synthesis (Ho et al., 2020), protein and material design (Corso et al., 2023), and probabilistic forecasting (Price et al., 2025; Morel et al., 2025; Rozet et al., 2025). By learning a stochastic process from a training dataset, these models can generate arbitrarily many plausible samples conditioned on partial information. They are particularly useful in the physical sciences, where data is often scarce and multiple states may be consistent with available observations (Epstein & Fleming, 1971; Nathaniel & Gentine, 2025). While perceptual applications mainly aim for plausibility, scientific and engineering problems require samples that obey strict physical or structural constraints, such as conservation laws or system dynamics (Kashinath et al., 2021). In such cases, approximate resemblance is not enough: generated samples must obey the governing physical principles. This requirement becomes even more critical when generative models are used out-of-distribution or in an autoregressive fashion, where small violations can accumulate and severely degrade long-term accuracy (Pedersen et al., 2025). Developing constrained sampling methods applicable to pre-trained generative models in a zero-shot scenario (*i.e.*, without additional training) is therefore crucial.

Modern generative models, including energy-based, score-based, and diffusion models (Du & Mordatch, 2019; Song et al., 2020), typically rely on Langevin dynamics, where noisy gradient steps push the samples toward high-likelihood regions. Enforcing mathematical constraints during Langevin sampling remains a challenging problem. A natural idea is to project each iterate onto the constraint set, leading to projected Langevin dynamics (Bubeck et al., 2015; Durmus et al., 2019; Christopher et al., 2024). While these methods offer theoretical guarantees in convex settings, they tend to perform poorly when applied to non-convex constraints, which are common in physical

---

*Correspondance to Matthieu Blanke <matthieu.blanke@nyu.edu>

systems. In such cases, strict projections can cause the dynamics to become trapped in limited regions of the constraint set, hindering exploration and introducing significant sampling bias. Other approaches use soft constraint penalty functions such as the barrier method (Fishman et al., 2023) and diffusion guidance (Ho & Salimans, 2022; Meunier et al., 2025), requiring a differentiable constraint model. These methods encourage but do not enforce constraints, which is insufficient when strict satisfaction is crucial. To our knowledge, no existing approach achieves both strict constraint satisfaction and unbiased exploration.

**Contributions** We introduce a rigorous mathematical formalism to analyze the constrained sampling problem. Inspired by the variational formulation of Langevin dynamics and constrained optimization, we propose Constrained Alternated Split Augmented Langevin (CASAL), a novel sampling algorithm that bridges the gap between complex generative modeling and constrained sampling using variable splitting. Our method enforces hard constraints while preserving the exploration capability of Langevin dynamics. It ensures strict constraint satisfaction and benefits from convergence guarantees via duality analysis. We show that our approach generalizes to deep generative modeling and diffusion models. We demonstrate the effectiveness of CASAL on complex physically constrained sampling tasks, including data assimilation problems where maintaining physical invariants is key to reliable forecasting, and on non-convex feasibility problems in optimal control.

## 2 PROBLEM FORMULATION OF CONSTRAINED LANGEVIN SAMPLING

In this section, we provide a mathematical formulation of constrained sampling: given a generative model and a constraint set, our goal is to generate samples from the conditional distribution supported on the constraint set. Such constrained distributions arise in many applications where samples must strictly satisfy known physical laws. We adopt the framework of the Langevin Monte Carlo algorithm (Rossky et al., 1978), a foundation of modern generative modeling frameworks. The application to deep generative models is discussed in Section 3.3.

**Langevin Monte Carlo** Consider a target distribution with density $p(x) = \mathrm{e}^{-f(x)}/Z$ on $\mathbb{R}^d$, where $f(x)$ is a differentiable potential. Markov chain Monte Carlo methods design iterative algorithms producing samples $(x_t)$ whose distribution $q_t$ converges to $p$. Among them, the Langevin Monte Carlo algorithm plays a central role. It requires access to the gradient of the potential $\nabla f(x)$, also called the score function (Hyvärinen & Dayan, 2005), and performs noisy gradient descent updates with a step size $\tau$ as

$$x_{t+1} = x_t - \tau \nabla f(x_t) + \sqrt{2\tau} w_t, \qquad w_t \overset{\text{i.i.d.}}{\sim} \mathcal{N}(0, I_d). \tag{2.1}$$

Under standard assumptions, the chain converges to $p$, the target distribution (Durmus et al., 2019).

**Constrained target distribution** We now consider the case where the samples are known to satisfy hard constraints at sampling time, in the form of a bounded measurable set $\mathcal{C} \subset \mathbb{R}^d$, which models prior information such as physical conservation laws. The conditional density supported on $\mathcal{C}$ is

$$p_{\mathcal{C}}(x) := \frac{1}{Z_{\mathcal{C}}} \mathrm{e}^{-f(x)} \mathbb{1}_{\mathcal{C}}(x), \quad \forall x \in \mathbb{R}^d, \tag{2.2}$$

with $\mathbb{1}_{\mathcal{C}}$ the indicator function of $\mathcal{C}$ and $Z_{\mathcal{C}}$ a normalizing constant. Note that the conditional distribution (2.2) can be rewritten using a modified potential: $p_{\mathcal{C}}(x) := \mathrm{e}^{-f_{\mathcal{C}}(x)}/Z_{\mathcal{C}}$, with the constrained potential $f_{\mathcal{C}}(x) := f(x) + \chi_{\mathcal{C}}(x)$, defined with the characteristic function of $\mathcal{C}$

$$\chi_{\mathcal{C}}(x) := \begin{cases} 0 & \text{if} \quad x \in \mathcal{C}, \\ +\infty & \text{otherwise.} \end{cases} \tag{2.3}$$

We do not make any assumption on the constraint set $\mathcal{C}$, except that it is bounded and that $p_{\mathcal{C}}$ is well-defined. Next, we provide examples of such constraints that may occur in physical applications.

**Example 2.1** [Physical constraints] When $x$ describes a discretized physical field, conservation of energy $E$ can often be expressed as the non-convex set $\mathcal{C} = \{x \in \mathbb{R}^d \mid \|x\|_2^2 = E\}$, while mass conservation corresponds to $\mathcal{C} = \{x \in \mathbb{R}^d \mid \sum_i x_i = M\}$ for a prescribed mass $M$.

**Objective**  Our objective is to design a sampling algorithm that produces samples distributed according to $p_\mathcal{C}$ for any constraint set $\mathcal{C}$. It should rely on the score function $\nabla f(x)$ of the unconstrained density, and mathematical operations related to $\mathcal{C}$ such as constraint functions, or projection operators. The method should operate in a "zero-shot" scenario, requiring no retraining or additional data.

**Example 2.2** [Projected Langevin] A natural idea to enforce hard constraints is to project each unconstrained update (2.1) onto $\mathcal{C}$ with the projection operator $P_\mathcal{C}(x) := \underset{z \in \mathcal{C}}{\operatorname{argmin}} \|x - z\|^2$, as

$$x_{t+1} = P_\mathcal{C}(x_t - \tau \nabla f(x_t) + \sqrt{2\tau} w_t), \qquad w_t \overset{\text{i.i.d.}}{\sim} \mathcal{N}(0, I_d). \tag{2.4}$$

This projected Langevin algorithm, and its extension to diffusion models, enjoy strong theoretical guarantees when $\mathcal{C}$ is convex and $p$ is log-concave (Bubeck et al., 2015). However, with nonconvex constraints, repeated projection can trap the dynamics in small feasible regions, biasing exploration (Barber & Ha, 2018; Ahn & Chewi, 2021). This motivates the need for sampling methods that enforce constraints more gradually.

**Example 2.3** [Soft penalty methods] Constraints can also be enforced softly by adding a differentiable cost $c(x) \geq 0$ to the potential, penalizing samples far from $\mathcal{C}$, with a tunable coefficient $\lambda \in \mathbb{R}$:

$$x_{t+1} = x_t - \tau(\nabla f(x_t) + \lambda \nabla c(x_t)) + \sqrt{2\tau} w_t, \qquad w_t \overset{\text{i.i.d.}}{\sim} \mathcal{N}(0, I_d). \tag{2.5}$$

This corresponds to guidance in diffusion models (Ho & Salimans, 2022; Huang et al., 2024c). The cost function $c(x)$ corresponds to a negative log-likelihood centered on the constraint set. Such methods encourage constraint satisfaction but do not guarantee it, as violations are only smoothly penalized.

**Evaluation**  Assessing the performance of constrained sampling algorithms is difficult as $p_\mathcal{C}$ is generally intractable. In practice, we rely on three key performance criteria: constraint violation, sampling bias, and computational cost. Constraint violation measures the deviation of samples from $\mathcal{C}$. Even when samples lie within $\mathcal{C}$, they must accurately follow the conditional distribution $p_\mathcal{C}$ without bias; this is typically quantified by comparing sample statistics to known or approximated quantities under $p_\mathcal{C}$. Finally, the computational cost is evaluated based on the number of function evaluations or total runtime required to generate valid samples.

## 3  SPLIT-AUGMENTED LANGEVIN FOR STRICTLY CONSTRAINED SAMPLING

In this section, we introduce a rigorous framework for constrained generation. We first derive a variational formulation of the target distribution $p_\mathcal{C}$ and, using Lagrangian duality, demonstrate why standard dual methods fail to sample from it. Building on this insight, we propose Constrained Alternated Split Augmented Langevin (CASAL), a primal-dual sampling algorithm that samples from $p_\mathcal{C}$ while ensuring strict constraint satisfaction through a variable splitting scheme.

### 3.1  VARIATIONAL FORMULATION OF CONSTRAINED SAMPLING

To better understand the constrained sampling problem, we formulate it as an optimization problem in the space of probability measures, leveraging the connection between Langevin dynamics and optimization in Wasserstein space. We let $\mathcal{P}_2(\mathbb{R}^d)$ denote the set of probability measures on $\mathbb{R}^d$ with finite second moments. For a measure $q \in \mathcal{P}_2(\mathbb{R}^d)$ admitting a density, we identify the measure with its density function. Further details on this framework are provided in Appendix A.

**Variational view of sampling**  Langevin Monte Carlo can be viewed as an optimization algorithm in distribution space. Let

$$D(q\|p) = \int \log\left(\frac{\mathrm{d}q}{\mathrm{d}p}\right) \mathrm{d}q \tag{3.1}$$

the Kullback-Leibler divergence (Kullback & Leibler, 1951). It is a non-negative information-theoretic quantity measuring how $q$ differs from $p$. Crucially, it can be shown that Langevin dynamics describe the gradient flow minimizing the functional $q \mapsto D(q\|p)$ in the Wasserstein

space $\mathcal{P}_2(\mathbb{R}^d)$ (Jordan et al., 1998; Villani, 2021). Consequently, the discrete-time Langevin algorithm (2.1) acts as a stochastic particle approximation of gradient descent in $\mathcal{P}_2(\mathbb{R}^d)$, driving the distribution $q_t$ of the chain toward the minimizer $p$ (Bernton, 2018). Formally, Langevin sampling solves:

$$\underset{q \in \mathcal{P}_2(\mathbb{R}^d)}{\text{minimize}} \quad D(q\|p). \tag{3.2}$$

Our approach extends this variational perspective to the constrained setting by characterizing $p_\mathcal{C}$ as the solution to a constrained optimization problem.

**Proposition 3.1** [Information projection] Suppose that $0 < \mathbb{P}_p(\mathcal{C}) < 1$. Then the conditional distribution $p_\mathcal{C}$ is the projection of $p$ onto the set of distributions supported on $\mathcal{C}$:

$$\begin{aligned} p_\mathcal{C} = \underset{q \in \mathcal{P}_2(\mathbb{R}^d)}{\text{argmin}} \quad & D(q\|p) \\ \text{subject to} \quad & \mathbb{P}_q(x \in \mathcal{C}) = 1. \end{aligned} \tag{3.3}$$

This is a special case of I-projection (Csiszár, 1975). To solve it, one might try to apply Lagrangian duality on (3.3). However, we show in the following result that strong duality is not attained, implying that duality-based numerical methods would fail to converge to $p_\mathcal{C}$.

**Proposition 3.2** Strong duality is not attained for (3.3).

We also show that penalty methods of Example 2.3 approximate (3.3) with a finite penalty parameter. Using Proposition 3.2, we prove that such methods cannot sample from $p_\mathcal{C}$.

**Corollary 1** Penalty methods (2.5) cannot enforce $\mathbb{P}_q(\mathcal{C}) = 1$.

This singularity arises because the support of the target distribution is a strict subset of $\mathbb{R}^d$, which prevents the constraint qualification from being satisfied. A possible relaxation is to allow a small violation probability $\mathbb{P}_q(\mathcal{C}) \geq 1 - \delta$ for small $\delta > 0$, but this allows unphysical states and leads to poor conditioning. To overcome this, we introduce a splitting strategy that relaxes the coupling between the sample and the constraint while maintaining strict feasibility.

## 3.2 Constrained Alternated Split Augmented Langevin

Directly targeting $p_\mathcal{C}$ is challenging because the potential forces exploration in $\mathbb{R}^d$ while the constraint forces the measure onto a lower-dimensional manifold. To address this composite objective, we propose to split the variable $x$ into a pair $(x, z) \in \mathbb{R}^d \times \mathcal{C}$, enforcing that $z \in \mathcal{C}$ while encouraging $x$ and $z$ to remain close. We thus define a joint probability measure $q(x, z)$, with marginals $q^x$ and $q^z$.

**Proposition 3.3** [Variable splitting] Problem (3.3) is equivalent to the following problem:

$$\begin{aligned} \underset{q \in \mathcal{P}_2(\mathbb{R}^d \times \mathbb{R}^d)}{\text{minimize}} \quad & D(q^x\|p) + \mathbb{E}_{q^z}\left[\chi_\mathcal{C}(z)\right] \\ \text{subject to} \quad & \mathbb{P}_q(x = z) = 1. \end{aligned} \tag{3.4}$$

This formulation mirrors variable splitting techniques in optimization (Gabay & Mercier, 1976), and separates the roles of $x$ and $z \in \mathcal{C}$, which are respectively maximizing likelihood and enforcing the constraint. Rather than requiring $x = z$ almost surely, we relax the condition to be satisfied in expectation, and penalize the variance. Specifically, we define a penalty parameter $\rho > 0$ and consider the following problem and its associated Lagrangian.

$$\begin{aligned} \underset{q \in \mathcal{P}_2(\mathbb{R}^d \times \mathbb{R}^d)}{\text{minimize}} \quad & D(q^x\|p) + \mathbb{E}\left[\chi_\mathcal{C}(z)\right] + \frac{\rho}{2}\mathbb{E}_q\left[\|x - z\|^2\right] \\ \text{subject to} \quad & \mathbb{E}_q[x - z] = 0. \end{aligned} \tag{P}$$

**Definition 1** For a primal-dual pair $(q, \lambda) \in \mathcal{P}_2(\mathbb{R}^d \times \mathbb{R}^d) \times \mathbb{R}^d$, the Lagrangian of (P) is

$$L(q, \lambda) := D(q^x\|p) + \mathbb{E}_{q^z}\left[\chi_\mathcal{C}(z)\right] + \lambda^\top \mathbb{E}_q[x - z] + \frac{\rho}{2}\mathbb{E}_q[\|x - z\|^2]. \tag{3.5}$$

Unlike the original projection Problem (3.3), this relaxed formulation admits a qualified constraint, ensuring that strong duality holds.

**Proposition 3.4** [Strong duality] Strong duality holds and is attained for (P). In particular, the Lagrangian (3.5) admits a saddle point $(q_\star, \lambda_\star)$, and $q_\star$ is a solution of (P).

Proposition 3.4 ensures that a solution of the relaxed problem (P) can be found by searching for a saddle point of (3.5). The primal density $q \in \mathcal{P}_2(\mathbb{R}^d \times \mathbb{R}^d)$ is approximated with particles.

**Stochastic saddle point iterations**    We propose a stochastic primal-dual scheme to solve the saddle point problem for (3.5). We generalize the method of Chamon et al. (2024) to our non-smooth, split setting. Given step size $\tau$ and noise $w_t \sim \mathcal{N}(0, I_d)$, the iterations are:

$$x_{t+1} = x_t - \tau \nabla f(x_t) - \tau\rho(x_t - z_t + \mu_t) + \sqrt{2\tau}w_t \tag{3.6a}$$
$$z_{t+1} = P_{\mathcal{C}}(z_t - \tau\rho(z_t - x_{t+1} - \mu_t)) \tag{3.6b}$$
$$\mu_{t+1} = \mu_t + (\tau/\rho)(x_{t+1} - z_{t+1}), \tag{3.6c}$$

where $\mu := (1/\rho) \times \lambda$ is a rescaled dual variable. This scheme, summarized in Algorithm 1, is named Constrained Alternated Split Augmented Langevin (CASAL). The primal variables $x$ and $z$ approximate the Wasserstein gradients of (3.5) with a gradient step and a proximal step respectively. In particular, $x$ follows Langevin dynamics biased toward $z$, while $z$ is projected onto $\mathcal{C}$. The dual variable $\mu$ integrates the error to correct bias, following a stochastic approximation of the dual ascent step.

---

**Algorithm 1** Constrained Alternated Split Augmented Langevin (CASAL)

> **input** potential gradient $\nabla f$, projection $P_{\mathcal{C}}$, step size $\tau > 0$,
> coupling $\rho > 0$, iteration number $T$, initial distribution $q_0$
> **output** sample $z_T \in \mathcal{C}$
> **initialize** $x_0 \sim q_0$, $z_0 = P_{\mathcal{C}}(x_0)$, $\mu_0 \in \mathbb{R}^d$
> **for** $0 \leq t \leq T - 1$ **do**
>     draw $w_t \sim \mathcal{N}(0, I_d)$
>     $x_{t+1} = x_t - \tau \nabla f(x_t) - \tau\rho(x_t - z_t + \mu_t) + \sqrt{2\tau}w_t$
>     $z_{t+1} = P_{\mathcal{C}}(z_t - \tau\rho(z_t - x_{t+1} - \mu_t))$
>     $\mu_{t+1} = \mu_t + (\tau/\rho) \times (x_{t+1} - z_{t+1})$
> **end for**

---

**Connection with optimization algorithms**    The update formulas (3.6) resemble the Alternating Direction Method of Multipliers (ADMM) of Glowinski & Marroco (1975) and Gabay & Mercier (1976), widely used in constrained optimization (Boyd et al., 2011). Here, the variables $x$ and $z$ play the role of the primal variables in ADMM and $\lambda$ the dual. Our sampling scheme can be seen a stochastic analog of the proximal ADMM (He et al., 2002) in sample space $\mathbb{R}^d$, just like Langevin Monte Carlo parallels gradient descent. However, it differs from ADMM applied in distribution space $\mathcal{P}_2(\mathbb{R}^d)$, as our method operates directly on coupled samples.

### 3.3 Practical implementation and deep generative models

**Implementation in diffusion models**    Our proposed algorithm is a constrained variant of Langevin Monte Carlo, which plays a central role in many generative frameworks (Du & Mordatch, 2019; Song & Ermon, 2019). The split-augmented updates (3.6) can be used as a drop-in replacement for standard Langevin steps, without altering other sampler components, making constraint enforcement simple and modular. Leveraging the connection between Langevin dynamics and diffusion models (Ho et al., 2020), CASAL provides a training-free constrained sampling algorithm for pretrained diffusion models. This parallel has already been exploited by Christopher et al. (2024) to introduce projected diffusion models. Details are discussed in Appendix B.

**Constraint satisfaction**    Our algorithm applies to arbitrary constraint sets, provided that a projection operator (exact or approximate) is available. Unlike penalty and latent projected methods, it does not require a differentiable constraint model, which can be challenging to derive (Laumond, 1987). The coupling parameter $\rho$ can be tuned or progressively increased along the diffusion process. This is detailed with ablation studies in Appendix C.

**Latent space constraints** A significant challenge arises when sampling occurs in a latent space $\mathbb{R}^d$, while the constraint set $\mathcal{C}$ is defined in a distinct physical space $\mathbb{R}^k$, linked by a decoder $A \in \mathbb{R}^{k \times d}$. The target density then takes the form:

$$p_{\mathcal{C}}(x) = \frac{1}{Z_{\mathcal{C}}} e^{-f(x)} \mathbb{1}_{\mathcal{C}}(Ax), \quad \forall x \in \mathbb{R}^d. \tag{3.7}$$

Enforcing constraints in this setting is significantly harder, as mapping a projection or penalty gradient from $\mathcal{C} \subset \mathbb{R}^k$ back to $\mathbb{R}^d$ implicitly requires inverting $A$, which is often computationally expensive or unstable due to ill-conditioning. In contrast, our framework naturally accommodates this setting by modifying the splitting constraint to $Ax = z$, instead of $x = z$. Crucially, this allows CASAL to perform the projection step solely in the physical space, without the need to invert the decoder or define a projection in the latent space.

**Computational cost** For learning methods to accelerate large-scale physical simulations, efficiency is central. Compared to unconstrained diffusion, our method adds the cost of a projection operation at each step, as does projected diffusion. For non-convex constraints, efficient numerical methods such as augmented Lagrangian algorithms can be used to solve the projection step, and are amenable to parallelization (Boyd et al., 2011; Liang et al., 2025). In the case of latent diffusion, our splitting technique allows the constraint operations to be computed directly in physical space, which can be considerably faster than propagating them through a decoder as in projected diffusion or diffusion guidance. More details can be found in Appendix C.

## 4 NON-ASYMPTOTIC CONVERGENCE ANALYSIS

We now provide a theoretical analysis of CASAL. Our study is twofold: first, we quantify the relaxation error induced by approximating the strict projection with the variable splitting problem (P). Second, we analyze the algorithmic convergence of the discrete-time process (3.6) toward the solution of the relaxed problem. Proofs are deferred to Appendix D.

### 4.1 PROPERTIES OF THE RELAXED SOLUTION

Let $(q_\star, \lambda_\star)$ denote a saddle point of the Lagrangian (3.5). We first characterize the optimum $q_\star$.

**Proposition 4.1** The optimal distribution $q_\star^x$ takes the form

$$q_\star^x(x) \propto \exp\left(-f(x) - \frac{\rho}{2} d_{\mathcal{C}}^2(x + \mu_\star)\right), \tag{4.1}$$

where $\mu_\star = (1/\rho) \times \lambda_\star$ and $d_{\mathcal{C}}(z) = \min_{y \in \mathcal{C}} \|y - z\|$ is the distance to the constraint set.

This result illustrates the mechanism of CASAL: the dual variable $\mu_\star$ shifts the effective potential to center the distribution correctly relative to the constraint, while $\rho$ controls the tightness of the constraint enforcement. In the limit of infinite penalty, we recover the exact conditional distribution.

**Proposition 4.2** [Recovery of the projection] Let $q_\rho$ denote the solution to (P). Then

$$q_\rho^x, q_\rho^z \underset{\rho \to +\infty}{\longrightarrow} p_{\mathcal{C}} \quad \text{in distribution.} \tag{4.2}$$

Thus, larger values of $\rho$ bring the $x$ samples closer to $\mathcal{C}$, while smaller values encourage exploration. For finite $\rho$, the following bounds quantify the approximation error relative to $p_{\mathcal{C}}$.

**Proposition 4.3** [Relaxation error] The relaxed solution $q_\star$ satisfies the following conditions:

$$D(q_\star^x \| p) \leq D(p_{\mathcal{C}} \| p), \quad \mathbb{P}_{q_\star^z}(\mathcal{C}) = 1, \quad W_2^2(q_\star^x, q_\star^z) \leq \frac{1}{\rho} D(p_{\mathcal{C}} \| p). \tag{4.3}$$

In practice, $\rho$ is finite, and Proposition 4.1 suggests a bias for finite $\rho$. The role of the Lagrange multiplier $\lambda_\star$ is to correct this bias, and center the density towards the right mode.

**Proposition 4.4** [Consistency] In the case of a non-degenerate Gaussian potential $f$ and an affine constraint set $\mathcal{C}$, the solution of (P) is unbiased: $\mathbb{E}_{q_\star}[x] = \mathbb{E}_{q_\star}[z] = \mathbb{E}_{p_{\mathcal{C}}}[x]$.

This result confirms the importance of using duality. As it is classical in optimization, duality allows for convergence to the constrained optimum with finite penalty parameter (Bertsekas, 2014).

## 4.2 CONVERGENCE AND MIXING RATE

We now analyze the convergence of CASAL to the stationary distribution $q_\star$. We rely on the following standard assumptions.

**Assumptions** The constraint set $\mathcal{C}$ is convex and bounded. The potential $f$ is $\alpha$-convex and $\beta$-smooth. Furthermore, there exists $M > 0$ such that for all $t$, $\|x_t - z_t + \mu_t\| \leq M$.

We define a Lyapunov function measuring convergence in primal and dual spaces.

**Definition 2** [Lyapunov function] For a primal-dual pair $(q, \lambda) \in \mathcal{P}_2(\mathbb{R}^d \times \mathbb{R}^d) \times \mathbb{R}^d$, we define the following Lyapunov function with respect to a reference pair $(p, \nu) \in \mathcal{P}_2(\mathbb{R}^d) \times \mathbb{R}^d$:

$$V(q, \lambda; p, \nu) := \frac{1}{2} W_2^2(q^x, p^x) + \frac{1}{2} W_2^2(q^z, p^z) + \frac{1}{2}\|\lambda - \nu\|^2. \tag{4.4}$$

The following proposition establishes a discrete evolution variational inequality (Pang & Stewart, 2008) for the algorithm. In particular, it gives a condition on the algorithm's hyperparameters $\rho$ and $\tau$ to ensure a decrease in the Lyapunov function.

**Proposition 4.5** [Primal-dual Lyapunov decrement] Let $q_+$ be the distribution after iterations (3.6) with step size $\tau$. Then, provided $4\tau(\beta + 4\rho) \leq 1$, for any reference pair $(p, \nu)$:

$$V(q, \lambda; p, \nu) - \mathbb{E}[V(q_+, \lambda_+; p, \nu)|\lambda] + \frac{\alpha}{2} W_2^2(q, q_+) \geq \tau(L(q_+, \nu) - L(q_\star, \lambda)) + C\tau^2. \tag{4.5}$$

Summing this decrement over $T$ iterations yields a convergence rate for the time-averaged distribution, matching the standard rates for Langevin Monte Carlo (Durmus et al., 2019).

**Corollary 2** [Mixing rate] Let $(x_t)$ be generated by Algorithm 1 with step size $\tau_t := 1/\sqrt{t+1}$. Let $\bar{q}_t$ denote the distribution of the time-averaged iterate $\bar{x}_t := \frac{1}{t}\sum_{s=0}^{t-1} x_s$. Then,

$$D(\bar{q}_t \| q_\star^x) \leq \mathcal{O}\left(\frac{\ln t}{\sqrt{t}}\right). \tag{4.6}$$

## 5 APPLICATION TO PHYSICALLY CONSTRAINED GENERATIVE MODELING

We evaluate CASAL on three scientific generative modeling tasks where challenging non-convex physical constraints play a critical role: stationary distribution sampling, data assimilation and optimal control. We apply our algorithm to diffusion models as described in Section 3.3. More details are given in Appendix C. Although it is not the central contribution of our experimental work, we also test CASAL on high-dimensional partial differential equations in Appendix C.5.

**Baselines** Our sampling algorithm is compared with the unconstrained Langevin algorithm, the projected Langevin algorithm, constraint penalty guidance methods, and their deep diffusion model analogs (Carvalho et al., 2023; Huang et al., 2024a; Christopher et al., 2024; Zampini et al., 2025). All methods share the same score function, and differ only in how constraints are incorporated.

### 5.1 ENERGY-CONSTRAINED STATIONARY FIELD GENERATION

We first validate our method on constrained Monte Carlo sampling of a stationary distribution, which is a critical problem in climate science and in molecular dynamics for example (Paquet & Viktor, 2015; Pedersen et al., 2025). We consider a two-dimensional field, representing for instance a fluid (see Figure 1), discretized on a $100 \times 100$ grid. The equilibrium distribution $p$ is sampled using Langevin dynamics.

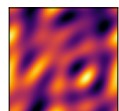

Figure 1: Sampled field snapshot.

**Constraint** A key macroscopic quantity is the kinetic energy, which often remains conserved or decreases in physical prediction tasks. Our task is to sample from the conditional distribution under a fixed energy $\mathcal{C} = \{x \in \mathbb{R}^d \mid \frac{1}{2}\|x\|_2^2 = E\}$, a non-convex constraint.

**Experimental setup**   The unconstrained distribution $p$ is bimodal in Fourier space, with asymmetric modes on the first Fourier coefficient: one positive and concentrated, the other negative and wider, allowing higher energy. The unconstrained distribution is sampled with the Langevin Monte Carlo algorithm, and $p_{\mathcal{C}}$ is estimated via rejection sampling. The bimodal nature of $p$ makes the exploration challenging. We condition on a high energy level, only achievable via the negative mode. As the positive mode cannot satisfy the energy constraint, the correct conditional distribution concentrates on the negative mode, and we can easily compare it to the generated samples. For each method, 1000 independent chains are run and the last iterate is collected. We compute histograms of the first Fourier coefficient for evaluation.

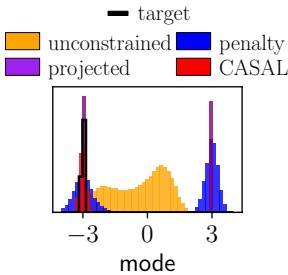

Figure 2: Empirical histograms for the first mode.

**Results**   Our results are shown in Figure 2. Only CASAL matches $p_{\mathcal{C}}$ closely. Projected Langevin satisfies the constraint exactly but fails to explore, yielding many samples in the wrong mode. Soft constraint penalty (Zhang et al., 2025) enforces energy conservation only on average, and therefore the produced samples do not match $p_{\mathcal{C}}$. These results demonstrate that CASAL enforces hard constraints while retaining enough exploration to correctly sample the conditional distribution.

## 5.2 Physics-preserving data assimilation

Data assimilation, a central problem in geophysics, aims to estimate the state of a dynamical system from sparse, noisy observations using prior knowledge. Recent work applies deep generative architectures to this task (Rozet & Louppe, 2023; Qu et al., 2024), but these models do not enforce physical invariants, such as energy or mass conservation, which are essential for physical plausibility in long-term forecasting. We study physically constrained generative models for data assimilation on the inviscid Burgers equation, a reduction of the Navier-Stokes equations with conserved mass and energy that exhibits rich dynamics and complex multiscale behaviors similar to turbulence (van Gastelen et al., 2024). A latent diffusion model is trained offline on a dataset of trajectories in Fourier space, without any conditioning. Appendix C.3 gives additional background.

**Constraint**   For the inviscid Burgers equation, states must satisfy mass and energy conservation, expressed in physical space $\mathbb{R}^k$ as $\mathcal{C} = \left\{ z \in \mathbb{R}^k \mid \|z\|_2^2 = E, \sum_i z_i = M \right\}$. This constraint set is non-convex, and the projection onto $\mathcal{C}$ is challenging to compute. The projection operator $P_{\mathcal{C}}$ is approximated via an iterative alternating-projection algorithm in high dimension. It is even more challenging to compute in the latent Fourier space, where $p_{\mathcal{C}}$ takes the form (3.7). More details are provided in Appendix C.3.

**Experimental setup**   We perform cyclic data assimilation on the Burgers equation discretized on a 200-point spatial grid. The ground truth trajectory evolves from a random initial condition over a time horizon $H = 8$. Observations are sparse: the system is observed at 10 equally spaced times, with 4 noisy spatial measurements at fixed, evenly spaced locations. Each method runs for 5 cycles per trajectory, producing a predicted trajectory that can be compared to the ground truth. The first baseline is 3D-Var (Courtier et al., 1998), which estimates the state with a Gaussian posterior. At sampling time, the diffusion model is combined with the Gaussian posterior, which conditions sampling to the available information. For each cycle, the analysis is computed as the average of 5 diffusion posterior samples. The experiment is repeated over 50 independent trajectories. We compute the average mean squared error with respect to the ground truth in the state space, in $\ell_2$ norm, and in the constraint space, where the quadratic constraint violation error is reported. All methods share the same biased linear forecast model.

**Results**   Figure 3 shows assimilated states and averaged error curves. In this under-observed setting, the diffusion prior helps to regularize the structure of complex states better than the Gaussian prior, especially for longer times, where the system shows a stiffer structure. However, unconstrained diffusion drifts away from the true trajectory, with significant deviations in both mass and energy. Projected diffusion (Christopher et al., 2024) strictly enforces constraints but introduces high-frequency artifacts, leading to physically implausible states. Our algorithm CASAL achieves the best compromise: it respects conservation and guides sampling toward physically plausible

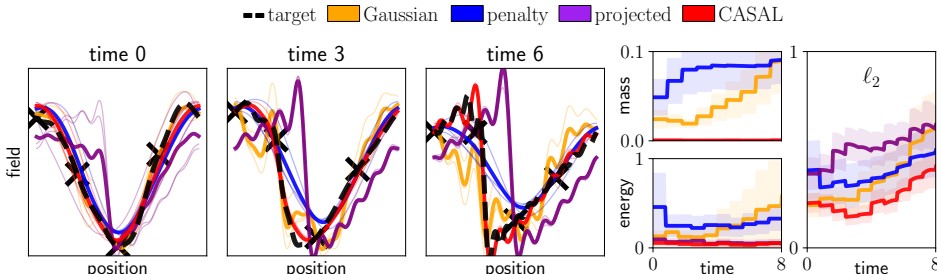

Figure 3: **Left** Data assimilation sampled states and reanalysis. The black crosses represent the observations. **Right** Averaged relative error, in terms of constraint violation and $\ell_2$ norm.

states, resulting in significantly lower estimation error. Wall-clock computational times are reported in Appendix C.3, and show the efficiency of our variable splitting approach in constraining latent diffusion model. These results highlight the potential of constrained generative modeling for robust data assimilation in physical systems.

### 5.3 CONSTRAINED PRIORS FOR FEASIBILITY PROBLEMS IN OPTIMAL CONTROL

As a final application, we evaluate CASAL on a feasibility problem in optimal control: find trajectories that satisfy both system dynamics and non-convex obstacle avoidance constraints. These problems are hard due to the non-convexity of obstacle regions. We consider a dynamical system with state $y(s)$ and control $u(s)$, with $s$ the physical time, and define a trajectory as $x := (y(s), u(s))_s$.

**Constraint**  Dynamics are encoded via the constraint set $\mathcal{C}_d := \{x \mid \dot{y} = F(y, u), \ |u| \leq u_{\max}\}$. Obstacle constraints define the potentially non-convex set $\mathcal{C}_o := \{x \mid y(s) \notin O_i \ \forall s\}$, for obstacle regions $O_i$. The goal is to find trajectories in the intersection $\mathcal{C}_d \cap \mathcal{C}_o$. For this task, ADMM is a classical solver alternating projections onto $\mathcal{C}_d$ and $\mathcal{C}_o$, but its convergence can be compromised when $\mathcal{C}_o$ is non-convex. Instead, we propose to guide ADMM with samples from a generative prior: a diffusion model trained on trajectories, with constraints enforced at sampling. This approach has seen promising results in control and robotics with diffusion penalty guidance and projected diffusion (Carvalho et al., 2023; Shaoul et al., 2025; Zampini et al., 2025), which we implement and compare with CASAL.

**Experimental setup**  We consider a planar quadrotor system, controlled in acceleration angle (Tedrake, 2009). A latent diffusion model is trained on a dataset of obstacle-free trajectories, obtained with a variety of random periodic excitations. At test time, non-convex obstacles are introduced. The corresponding constraint is imposed during sampling. In order to avoid the obstacles, the algorithm needs to find a swinging trajectory. Each sampled trajectory is then used to initialize an ADMM solver for feasibility problems (Bílková & Šorel, 2021), and we record the fraction of samples for which a feasible solution is found.

**Results**  Figure 4 shows some sampled trajectories and success rates as the obstacle size $r$ increases, computed over 10000 samples. Constraint penalty guidance favors obstacle avoidance, but some sampled trajectories penetrate the obstacles. Projected diffusion avoids obstacles but suffers projection bias, producing distorted and unphysical paths. Our algorithm balances both aspects: it produces obstacle-avoiding trajectories that remain dynamically feasible, leading to significantly higher success.

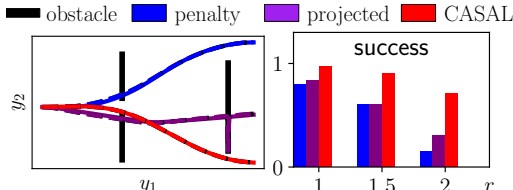

Figure 4: **Left** Dashed lines are sampled trajectories, solid lines are the projections onto the feasibility set. **Right** Success rates for various obstacle sizes of the rightmost obstacle.

## 6 RELATED WORK

**Constrained diffusion** Early approaches adapted optimization methods to Langevin dynamics, including Projected Langevin Monte Carlo (Bubeck et al., 2015), proximal Monte Carlo (Salim et al., 2019), Mirrored Langevin (Hsieh et al., 2018), and penalized Langevin (Gurbuzbalaban et al., 2024). Extensions to diffusion models have also been explored (Fishman et al., 2023; Liu et al., 2023; Christopher et al., 2024). These methods offer convergence guarantees in convex settings, where constraints do not hinder exploration, but are less effective in non-convex physical problems. Carvalho et al. (2023), Huang et al. (2024a) and Meunier et al. (2025) impose soft constraint penalties in the diffusion process. Crucially, these methods rely on a differentiable constraint penalty function. Therefore, the sampling objective is inherently different from (3.3), as the almost-sure constraint defines a non-smooth potential. Our approach tackles this non-smoothness using a proximal operator and projections, thereby ensuring strict constraint satisfaction, rather gradient steps on a smooth approached loss. The variational formulation of Langevin sampling has been used by Chamon et al. (2024) to enforce constraints on average.

**Variable splitting** Variable splitting, inspired by ADMM, has been applied to Bayesian posterior sampling (Vono et al., 2019), plug-and-play samplers for inverse problems (Bouman & Buzzard, 2023; Wu et al., 2024; Martin et al., 2024), and guided diffusion (Zhang et al., 2025). These works apply variable splitting to a smooth maximum a posteriori optimization problem, where the auxiliary variable is updated by gradient descent. Crucially, our framework enforces exact constraint satisfaction through non-smooth constraint potential, without requiring a differentiable constraint model. Moreover, we formalize sampling as an optimization problem in density space, which is key to obtain our probabilistic sampling guarantees. Our algorithm also extends to latent diffusion, enabling computational savings.

**Physically constrained neural networks** Physical constraints have also been imposed on deterministic neural networks (Négiar et al., 2023; Hansen et al., 2023). In related sampling approaches, Cheng et al. (2024) and Utkarsh et al. (2025) integrate projection into flow-matching. Our approach differs in targeting strict satisfaction in a diffusion framework.

## 7 CONCLUSION

In this work, we addressed the complex problem of constrained sampling from a novel theoretical perspective. While prior methods often rely on heuristics, we introduced the first mathematical framework for this task based on Lagrangian duality in the space of probability measures. This formalism not only provides a rigorous foundation but also offers a diagnostic tool to understand the limitations of projected and penalty-based approaches.

Our proposed algorithm, CASAL, offers a principled bridge between constrained sampling theory and modern generative modeling. By leveraging a variable splitting scheme, we ensure strict constraint satisfaction while maintaining the exploration properties of Langevin dynamics. The relaxation error and the mixing times are analytically quantified in Wasserstein distance. Crucially, our framework naturally extends to latent diffusion, where it remarkably decouples the physical constraint space from the generative latent space. Our applications to complex physical systems illustrate the practical importance of informing data-driven methods with physical constraints.

While CASAL naturally extends to deep generative models, several promising directions remain. A natural extension would be the derivation of a natively time-dependent consrtained variational formulation of the sampling problem for diffusion models and other modern deep generative models such as stochastic interpolants (Albergo & Vanden-Eijnden, 2023). Finally, while the reliance on a projection operator is the necessary price for strict feasibility, further research into approximate or parallelized projections could broaden the applicability of CASAL to more high-dimensional, constrained sampling problems in science and engineering.

**Reproducibility statement** The proofs of the new theoretical results included in this paper are available in Appendix D. The code of the proposed algorithm is available online at

https://github.com/MB-29/constrained-sampling,

and

https://github.com/MB-29/pascal.

Implementation details and comparison with other algorithms and diffusion models are discussed in Appendix B. Ablation studies, discussion about algorithm hyperparameters and experimental details are available in Appendix C.

**Acknowledgments** We thank Carla Roesch, Luiz Chamon and Anna Korba for their insightful feedback on this work. The authors acknowledge funding, computing, and storage resources from the NSF Science and Technology Center (STC) Learning the Earth with Artificial Intelligence and Physics (LEAP) (Award #2019625). The authors acknowledge the M²LInES organization for its support.

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

# A    VARIATIONAL FRAMEWORK FOR LANGEVIN MONTE CARLO

Consider the functional

$$F(q) = D(q\|p) = \int q \log(q/p). \tag{A.1}$$

The Wasserstein gradient flow is defined as the following differential system

$$\frac{\partial q}{\partial t} = \nabla \cdot \left( q \nabla \frac{\partial F}{\partial q} \right), \tag{A.2}$$

For functional (A.1), this differential system becomes

$$\frac{\partial q}{\partial t} = \nabla \cdot (q \nabla f(x)) + \Delta q(x, t), \tag{A.3}$$

which is found to be the Fokker-Planck equation for the Langevin dynamics

$$\mathrm{d}x = -\nabla f(x)\mathrm{d}t + \mathrm{d}B. \tag{A.4}$$

More details can be found in (Jordan et al., 1998; Ambrosio et al., 2008; Villani, 2021; Chamon et al., 2024).

## B    DEEP GENERATIVE MODELS AND LANGEVIN SAMPLING

Many modern generative frameworks, from energy-based models to state-of-the-art diffusion models, rely on Langevin dynamics for sampling (Hinton, 2002; Du & Mordatch, 2019; Song & Ermon, 2019; Song et al., 2020). In Appendix B, we review how key classes of generative models relate to Langevin updates.

For these generative models, sampling takes the form

$$x_{t+1} = x_t - \tau_t \nabla f(x_t, t) + \sqrt{2\tau_t}\, w_t, \quad w_t \sim \mathcal{N}(0, I). \tag{B.1}$$

We interpret these steps as the discretization of a Wasserstein flow for a time-dependent functional $F(q, t)$, which is summarized in Appendix A. We can then identically apply our constrained sampling algorithm, as a time-dependent variation of Algorithm 1, detailed in Algorithm 2. From a variational point of view, this results in framing the constrained sampling as a time-varying constrained optimization problem.

---

**Algorithm 2** Time-dependent CASAL

---

**input** time dependent potential gradient $\nabla f(x, t)$, iteration number $T$, time-dependent step sizes $\tau_t$, projection $P_{\mathcal{C}}$, coupling $\rho > 0$, intial distribution $q_0$
**output** sample $z_T \in \mathcal{C}$
**initialize** $x_0 \sim q_0$, $z_0 = P_{\mathcal{C}}(x_0)$, $\mu_0 = 0 \in \mathbb{R}^d$
**for** $0 \leq t \leq T - 1$ **do**
    draw $w_t \sim \mathcal{N}(0, I_d)$
    $x_{t+1} = x_t - \tau_t \nabla f(x_t, t) - \tau_t \rho(x_t - z_t + \mu_t) + \sqrt{2\tau_t}\, w_t$
    $z_{t+1} = P_{\mathcal{C}}(z_t - \tau_t \rho(z_t - x_{t+1} - \mu_t)$
    $\mu_{t+1} = \mu_t + \tau_t(x_{t+1} - z_{t+1})$
**end for**

---

### B.1    ENERGY-BASED MODELS

An energy-based model (EBM) defines a density

$$p(x) = \frac{1}{Z} \exp\big(-f_\theta(x)\big), \tag{B.2}$$

where $f_\theta$ is a learned energy function. Sampling from $p$ typically relies on Langevin dynamics (2.1) or stochastic gradient Langevin dynamics (Welling & Teh, 2011). Energy-based models with Langevin sampling have demonstrated strong performance across a range of tasks(Du & Mordatch, 2019), and offer distinct advantages over methods such as Variational Autoencoders (Kingma et al., 2013) and Generative Adversarial Networks (Goodfellow et al., 2014). A particularly valuable property of energy-based models is their flexibility in incorporating constraints via summing up the corresponding energies. From this perspective, our algorithm, when applied to energy-based models, can be interpreted as providing stronger constraint enforcement through an augmented Lagrangian potential and corresponding proximal Langevin updates—going beyond the simple addition of constraint energies.

### B.2    SCORE-BASED GENERATIVE MODELS

Score-based generative models aim to learn the score function $\nabla \log p_t(x)$ of a family of progressively noised data distributions $\{p_t\}_{t \in [0,T]}$, rather than modeling the data density directly. Once the score is learned, typically via denoising score matching, samples can be generated by Langevin-type updates.

Proposed by Song & Ermon (2019), this method generates samples by applying Langevin dynamics at a sequence of decreasing noise levels $\sigma_T > \cdots > \sigma_1$. A score model $s_\theta(x, \sigma)$ is trained to approximate the noise-dependent score $\nabla_x \log q(x; \sigma)$ of the perturbed data distribution $p(x; \sigma)$, which is obtained by convolving $p(x)$ with a Gaussian of various noise level $\sigma_t$. Then update step takes the form

$$x_{t+1} = x_t + \tau_t\, s_\theta(x, \sigma_t) + \sqrt{2\tau_t}\, w_t, \quad w_t \sim \mathcal{N}(0, I), \tag{B.3}$$

where $\tau_t \propto \sigma_t^2$ are time-varying step sizes. The update takes the form of (B.1) with $\nabla f(x, t) = -s_\theta(x, \sigma_t)$. This can be seen as an unadjusted Langevin algorithm with temperature $\sigma_t$, gradually refining the sample as noise decreases. In this case our algorithm can be directly applied at each noise level to impose constraints. It is worth noting that the projected diffusion model (Christopher et al., 2024) also falls into this category – a hard projection following each Langevin update within the annealed Langevin dynamics framework. Note that this covers the case where several Langevin steps are taken at fixed noise level, as in the work of Song & Ermon (2019), by choosing $\tau_t$ to be constant for a number of steps $t$.

## B.3 DIFFUSION MODELS

Denoising diffusion probabilistic models (DDPMs), introduced by Ho et al. (2020), define a forward process that gradually corrupts a data point $y_0$ by adding Gaussian noise through a fixed Markov chain:

$$q(y_t \mid y_{t-1}) = \mathcal{N}(y_t; \sqrt{1 - \beta_t}\, y_{t-1}, \beta_t I), \tag{B.4}$$

where $\beta_t \in (0, 1)$ is a small noise schedule. This leads to a closed-form expression for $q(x_t \mid x_0)$, with the following definitions:

$$\alpha_t = 1 - \beta_t, \quad \bar{\alpha}_t = \prod_{s=1}^{t} \alpha_s. \tag{B.5}$$

The reverse process is parameterized by a neural network $\epsilon_\theta(x_t, t)$, which predicts the noise component. The sampling procedure follows:

$$x_{t+1} = \frac{1}{\sqrt{\alpha_t}} \left( x_t - \frac{1 - \alpha_t}{\sqrt{1 - \bar{\alpha}_t}} \epsilon_\theta(x_t, t) \right) + \sigma_t w, \quad w_t \sim \mathcal{N}(0, I), \tag{B.6}$$

where $\sigma_t$ is typically set to match the forward variance $\beta_t$. As noted by Ho et al. (2020), this step corresponds to an Euler-Maruyama discretization of a variant of Langevin dynamics, and the learned noise predictor $\epsilon_\theta$ implicitly estimates the score $\nabla \log p_t(x)$ up to a scaling factor. Hence, the sampling formula (B.6) takes the form (B.1) with $\tau_t = \sigma_t^2/2$ and

$$\nabla \log p_t(x_t) \approx s_\theta(x_t, t) = -\frac{1}{\sqrt{1 - \bar{\alpha}_t}} \epsilon_\theta(x_t, t). \tag{B.7}$$

The DDPM can be regarded as a discrete score-based model under the variance preserving stochastic differential equation (VP-SDE) interpretation (Song et al., 2020), and thus our CASAL sampling is valid for DDPM sampling.

## B.4 SCORE-BASED DIFFUSION MODELS

Score-based diffusion models (Song et al., 2020) directly learn the score function of perturbed data distributions and generate samples by simulating the reverse-time stochastic dynamics.

**Forward SDE** Define a forward Itô SDE that gradually adds noise to data $x_0 \sim p_{\text{data}}$:

$$dx = a(x, t)\, dt + b(t)\, dW_t, \tag{B.8}$$

where for the variance-preserving (VP) choice,

$$a(x, t) = -\tfrac{1}{2}\,\beta(t)\, x, \quad b(t) = \sqrt{\beta(t)}. \tag{B.9}$$

This yields marginal distributions $p_t(x)$ that interpolate between the data and near-Gaussian noise as $t$ increases.

**Reverse SDE** The time-reversed process follows

$$dx = \big[a(x, t) - b(t)^2 \, \nabla_x \log p_t(x)\big] dt + b(t)\, dW_t', \tag{B.10}$$

where $W_t'$ is a reverse-time Wiener process. A neural network $s_\theta(x, t)$ is trained by score matching to approximate $\nabla_x \log p(x, t)$.

**Predictor–Corrector sampling.** Once the score network is trained, our CASAL sampling is applicable. CASAL can also be integrated seamlessly into the predictor-corrector sampling scheme proposed by Song et al. (2020). The predictor-corrector sampler interleaves the following steps.

- Predictor: an Euler–Maruyama step of the reverse SDE,

$$x_{t+1} = x_t - \tau_t \big[ a(x_t, t) - b(t)^2 \, s_\theta(x_t, t) \big] + b(t) \sqrt{2\tau_t} w_t \quad w_t \sim \mathcal{N}(0, I_d). \quad \text{(B.11)}$$

- Corrector: a few steps of Langevin MCMC to refine samples,

$$x_{t+1} = x_t + \tau_t \, s_\theta(x_t, t) + \sqrt{2\tau_t} w_t, \quad w_t \sim \mathcal{N}(0, I_d). \quad \text{(B.12)}$$

Similar to the previous sections, these formulas take the form of (B.1), with different time-varying potential gradients $\nabla f(x, t)$ and step sizes.

**Summary**   Across EBMs, diffusion models, and hybrid schemes, the core sampling formula is an overdamped Langevin update, possibly annealed through noise scales. This makes our constrained sampling algorithm CASAL compatible with all these approaches as a zero-shot plug-in.

## C  EXPERIMENTAL DETAILS

### C.1  PARTICLE ON A CIRCLE

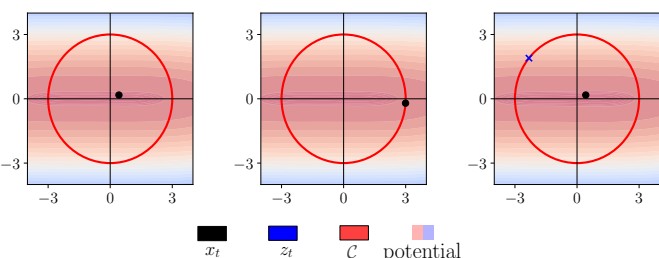

Figure 5: True posterior density, projected Langevin samples and CASAL.

Figure 5 shows the exploration issue arising with Projected Langevin Monte Carlo in the case of non-convex constraints and a bi-modal distribution. Here, projecting on the constraint set

$$\mathcal{C} = \{x \mid \|x\|^2 = E\} \tag{C.1}$$

leads to poor exploration, as the samples are stuck on the positive side of the likelihood landscape, while the only high-likelihood zone compatible with the constraint is on the other side. Our algorithm solves this exploration issue leveraging variable splitting

### C.2  FIELD GENERATION

**Baselines**  Our algorithm is compared to projected Langevin Monte Carlo, primal-dual Monte Carlo and constraint-penalized Langevin Monte Carlo. For the latter, we implement the variable-splitting algorithm of Zhang et al. (2025), and the penalty parameter is a dual variable that is adapted and updated following the same scheme as CASAL.

**Sampling**  Langevin Monte Carlo is iterated over 1000 steps, and we set $\rho$ to follow a linear interpolation schedule between 2 and 20.

**Constraints**  The field is subject to energy conservation (Example 2.1). The projection in closed forms. For the penalty method, the penalty cost is $c(x) = (\sum x_i - M)^2$. The primal-dual Langevin Monte Carlo algorithm enforces the constraint function $h(x) = \sum x_i - M$ on average.

### C.3  DATA ASSIMILATION

**Context**  In many geophysical and engineering applications, one relies on numerical simulation to predict the time-dependent evolution of a complex system, whose state at physical time $t$ is denoted by $x \in \mathbb{R}^d$. But these models are inherently imperfect—either because of computational constraints or incomplete knowledge of the true dynamics. When real-world observations $y \in \mathbb{R}^m$ become available (for example in digital-twin settings), we assume a statistical model of the form

$$y = h(x) + \varepsilon, \tag{C.2}$$

where $h : \mathbb{R}^d \to \mathbb{R}^m$ is an observation operator and $\varepsilon$ is the measurement error. The imperfect simulation yields a prior forecast $b \in \mathbb{R}^d$, the background estimate, which must be adjusted using $y$ to produce a more accurate estimate of the true state, usually referred to as the analysis, as the initial condition for the next simulation. Equivalently, one seeks samples from the posterior

$$p(x|b, y) \propto p(y|x)\, p(x|b). \tag{C.3}$$

This estimation problem is formulated sequentially for each new observation, by propagating the obtained posterior analysis with a forecast model, and repeating the process. Classically, this is achieved by one of three approaches: sequential Monte Carlo methods (e.g. particle filters (Gordon et al., 1993)), ensemble-based filters (e.g. the Ensemble Kalman Filter (Evensen, 2003)), or variational methods that solve for the MAP estimate (e.g. 3D-Var/4D-Var (Sasaki, 1970; Lorenc,

1986)). The 3D-Var algorithm assumes that the background error distribution and observation error distribution are Gaussian,

$$x|b \sim \mathcal{N}(b,\, B), \quad \varepsilon \sim \mathcal{N}(0,\, P), \tag{C.4}$$

then taking negative logarithm of (C.3) yields the following optimization target:

$$J(x) = \tfrac{1}{2}\big\|y - h(x)\big\|_{P^{-1}}^2 + \tfrac{1}{2}\big\|x - b\big\|_{B^{-1}}^2. \tag{C.5}$$

Deep learning represents a promising tool to learn more complex priors for data assimilation (Huang et al., 2024b; Rozet & Louppe, 2023; Qu et al., 2024; Blanke et al., 2024)

**Data**  For simulating the Burgers equation, we implemented the same method as van Gastelen et al. (2024), but we added an extra constant linear advection term. We work in Fourier space with the first 20 Fourier modes. The field evolves according to the Burgers equation for 4 time units. We generate 1,000 trajectories, with the field recorded at 10 timesteps for each trajectory, with the initial state drawn at random in Fourier space with a power-law decay of the coefficient magnitude.

**Learning architecture**  We implemented a DDPM diffusion model, using the formalism detailed in Appendix B. Diffusion is learned in a latent space, defined as the first 10 Fourier modes. The neural network involved is a fully connected network with depth 3 and width 128, using a cosine time embedding. It is trained for 200 epochs. At sampling time, 1000 diffusion steps are used with $\rho = 10$.

**Baselines**  Our algorithm is compared to unconstrained latent diffusion, latent penalty diffusion and latent projected diffusion, incorporating observations using diffusion posterior sampling Rout et al. (2023) or as a strict constraint, and propagating the projection and the penalties through the decoder.

**Constraints**  The field is sampled subject to energy and mass conservation constraints (Example 2.1). The projection is computed by alternating projections on the two constraint set, which have closed forms.

The initial conditions are drawn at random following the same distribution of the training data.

**Additional results**  Figure 6 shows the evolution of key estimation metrics for a data assimilation trajectory, for the various methods compared.

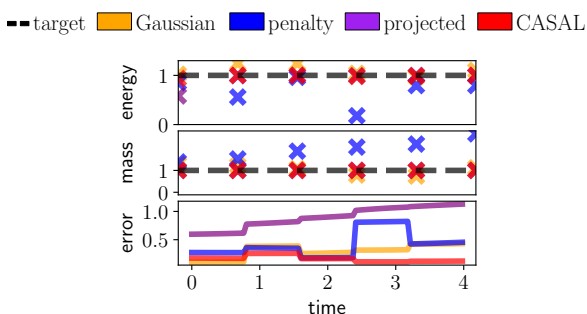

Figure 6: Mass conservation, energy conservation and $\ell_2$ error.

## C.4  FEASIBILITY PROBLEM

Trajectory planning for quadrotor obstacle avoidance is an imoprtant problem (Le Hellard et al., 2025). We implement a linearized version of the planar quadrotor dynamics described in (Tedrake, 2009).

**Data**  The trajectories are discretized in time as $(y_1, \ldots, y_S, u_1, \ldots u_S) \in \mathbb{R}^{2S}$, with $S = 200$ and a time interval $\Delta s = 0.01$. The dynamics constraint

$$\mathcal{C}_{\mathrm{d}} := \{x \mid y(0) = 0, \dot{y}(s) = Ay(s) + Bu(s), |u(s)| \leq u_{\max}\} \tag{C.6}$$

is described by a linear equality constraint, discretized into a linear system, and an inequality constraint on the control inputs. The projection on this convex constraint set is obtained by Dykstra's double projection algorithm (Bílková & Šorel, 2021), and is used within the ADMM solver.

**Learning architecture**   We implemented a DDPM diffusion model, using the formalism detailed in Appendix B. The trajectories signals are learned in a latent Fourier space encoding the first 10 modes of the input signal. The neural network involved is a fully connected network with depth 3 and width 128, using a cosine time embedding. It is trained for 200 epochs. At sampling time, 1000 diffusion steps are used with $\rho = 100$.

**Baselines**   We implement the latent projected diffusion algorithm (Zampini et al., 2025), and diffusion guidance with constraint penalties (Carvalho et al., 2023), and propagate the penalty function through the decoder.

**Constraint**   The obstacles are segments, and projecting onto the feasible region is performed by moving the penetrating trajectory portions trajectory either directly above or directly underneath the obstacle. For the penalty method, the constraint penalty is the quadratic distance between the trajectory and the obstacle, which is simple enough to differentiate through in this case.

## C.5   PHYSICALLY-CONSTRAINED EMULATION OF PARTIAL DIFFERENTIAL EQUATIONS

To confirm the applicability of CASAL to a wider range of high-dimensional PDEs with complex constraints, we evaluate our method on three additional systems: the Darcy flow, the Navier-Stokes equations, and the 2D Poisson equation, following the experiment setting of Huang et al. (2024a). All methods in these experiments share the same network architecture and observational data.

**Darcy Flow**   We experiment on the Darcy flow setting of DiffusionPDE, discretized on a $32 \times 32$ grid. We use latent diffusion in Fourier space and sample both the coefficient field and the solution field from 25 sparse observations. The samplers aim at enforcing the Darcy equation connecting the coefficients and the solution. The reconstructed fields are posterior averages over 10 samples.

As shown in Table 1, CASAL significantly reduces the reconstruction error compared to the baselines while satisfying the physical constraints to a high degree of precision. All approaches demonstrated similar wall-clock computational times.

Table 1: Reconstruction error and constraint violation for the Darcy flow experiment.

| Algorithm | Reconstruction error | Constraint violation |
|---|---|---|
| Projected diffusion | 22 % | $1 \times 10^{-5}$ % |
| Penalty | 23 % | 25 % |
| CASAL | 10 % | $2 \times 10^{-5}$ % |

**Navier-Stokes**   Following the approach of DiffusionPDE, we evaluate generative data assimilation on the Navier-Stokes equations. We constrain the generated fields to match sparse observations while simultaneously strictly enforcing the physical constraints of enstrophy and total circulation.

Table 2 demonstrates that CASAL achieves the lowest reconstruction error for the initial and terminal states, while strictly satisfying the physical constraints at a computational cost comparable to the baselines.

Table 2: Reconstruction error, constraint violation, and runtime for the Navier-Stokes experiment.

| Algorithm | Reconstruction error | Constraint violation | Wall-clock time (s) |
|---|---|---|---|
| Projected diffusion | 10 % | $10^{-5}$ % | 1.07 |
| Penalty | 11 % | 20 % | 1.00 |
| CASAL | 8 % | $10^{-5}$ % | 1.05 |

**Poisson Equation**   To provide a clear demonstration of transfer beyond the Burgers equation to a purely elliptic setting, we benchmark CASAL on the 2D Poisson equation. We consider a $100 \times 100$ discretization and train a single latent diffusion model on solutions with random boundary conditions

in Fourier space. At inference time, 10% of the grid points are observed, and the global mass is known.

All samplers enforce the mass constraint (in this setting, penalty-guided diffusion reduces to a latent analogue of DiffusionPDE). Each sampler produces 50 posterior samples. The results in Table 3 show that CASAL substantially improves the sampling accuracy over baseline methods while strictly enforcing the mass constraint.

Table 3: Reconstruction error, constraint violation, and runtime for the Poisson equation.

| Algorithm | Reconstruction error | Constraint violation | Wall-clock time (s) |
|---|---|---|---|
| Projected diffusion | $15\%$ | $2 \times 10^{-7}\%$ | 0.34 |
| Penalty | $14\%$ | $25\%$ | 0.33 |
| CASAL | $5\%$ | $2 \times 10^{-7}\%$ | 0.32 |

### C.6 COMPUTATIONAL COST

To assess the computational cost of CASAL, we summarize below the costs for the constraint sets used in our experiments. We provide below a breakdown of the projection cost for different constraint types, and compare them to the overall runtime.

In our experiments, we use three types of base projections:

- projection onto a sphere for energy conservation,
- projection onto intervals for obstacle avoidance,
- projection onto a linear subspace for mass conservation,

We also combine these projections in the case where the constraint is the intersection of such sets. All of these admit efficient implementations.

For the sphere and the interval, cost is $\mathcal{O}(d)$ with closed-form formulas.

For a linear subspace,

$$\mathcal{C} = \{x \mid Ax + b = 0\} \tag{C.7}$$

with $A \in \mathbb{R}^{m \times d}$. The projection is given by $P_\mathcal{C}(x) = x + A^\top (AA^\top)^{-1}(b - Ax)$, which requires a precomputed pseudoinverse at cost $\mathcal{O}(d^2 m)$ and a matrix-vector product at cost $\mathcal{O}(m)$, which remains small compared to neural network evaluations.

In all tested settings, the runtime overhead from projections was small compared to the cost of score evaluations in diffusion models. Furthermore, CASAL is compatible with approximate projections, allowing further savings. For more complex constraint sets, iterative solvers such as ADMM can be employed with a limited number of steps, trading accuracy for speed in early iterations, where perfect constraint enforcement is not yet required.

To validate this point, we conducted the following additional runtime experiment. We measure the average wall-clock time for the different sampling algorithms in the data assimilation problem, where each sampled state is projected on the intersection of 2 constraint sets: one for mass and one for energy. All times are in seconds per $10^6$ sampling steps, measured on an Apple M1 setup.

| Experiment | unconstrained Langevin | penalty | projected Langevin | CASAL |
|---|---|---|---|---|
| Fluid generation | 0.37 | 0.41 | 0.43 | 0.45 |
| Data assimilation | 0.17 | 0.25 | 2.46 | 0.31 |

Table 4: Comparison of computational times.

We also note that diffusion guidance can be computationally costly as it can require multiple penalty gradient steps per sampling step to enforce constraints. In the experiment 5.3, we found this method to be of the order of 3 times slower than CASAL and projected diffusion.

In summary, CASAL has comparable runtime to projected Langevin. For a number of use cases, its additional cost is modest, especially in the context of deep generative models where the computational budget is dominated by score evaluations. This cost can also be adjusted in practice, by computing approximate projections in the early steps of sampling.

## C.7 Ablation studies for $\rho$ and $\lambda$

For the field generation task of Section 5.1, we generate 1000 samples of energy-constrained fields, using different schedules for $\rho$. For each schedule, we evaluate the samples with the following measure of error: we report the proportion of samples that fall near the unlikely positive mode of the bimodal potential, implying that the sampled distribution deviates from the target

We experiment with 4 different schedules: two schedules use constant $\rho$ throughout the iterations. The two other schedules are linear and logarithmic interpolation between these two values. We run the experiment for different numbers of Langevin iterations. The results are reported in the following table.

| number of steps / schedule | constant $\rho = \rho_{\min}$ | constant $\rho = \rho_{\max}$ | linear | logarithmic |
|---|---|---|---|---|
| 1000 | 42.4% | 22% | 0.04% | 3.8% |
| 5000 | 44.7% | 2.8% | 0.001% | 0.0002% |

Table 5: Proportion of samples in the wrong mode for different schedules of $\rho$

We observe that allowing $\rho$ to vary across iterations substantially improves sample quality. With too small $\rho$, the deviation between $x$ and $z$ is too large. With too large $\rho$, the chain fails to explore the energy landscape. When the number of steps is limited, only annealed schedules manage to recover the correct mode. This experiment highlights the importance of adaptive schedules in practice.

In the other experiments, we found that the time-varying step size induce by diffusion models, which also scales $\rho$, was sufficient to balance exploration and constraint satisfaction

We conduct an ablation study on both the field generation experiment (Section 5.1) and the Burgers data assimilation task (Section 5.2) to investigate the influence of the initial value of the dual variables.

In the first experiment, the fields are sampled with the Langevin Monte Carlo algorithm, with fixed potential. In the second experiment, a diffusion model is used, so the score function is time-varying function throughout iterations.

We run our sampling algorithm with Gaussian initialization of $\lambda$, with different sizes $\sigma$. For each value of $\sigma$, we run 100 independent chains. For the flow sampling experiment, we report the maximum number of sampling steps required to converge. For the data assimilation experiment, because the dual problem changes over time, we do not evaluate the convergence of the dual variables. Instead, we report the average reconstruction accuracy.

| $\sigma$ | Steps to convergence | Reconstruction error |
|---|---|---|
| 0 | 10 | 0.56 |
| 1 | 200 | 0.56 |
| 5 | 800 | 0.56 |
| 10 | 1000 | 0.57 |
| 20 | 1600 | 0.58 |
| 50 | 2500 | 0.79 |
| 100 | N/A | 1.34 |

Table 6: Influence of $\lambda$

## D PROOFS

In the following proofs, if $q \in \mathcal{P}_2(\mathbb{R}^d)$ has a density, we identify $q$ and its probability density function.

### D.1 PROOF OF PROPOSITION 3.1

**Proposition 3.1** [Information projection] Suppose that $0 < \mathbb{P}_p(\mathcal{C}) < 1$. Then the conditional distribution $p_\mathcal{C}$ is the projection of $p$ onto the set of distributions supported on $\mathcal{C}$:

$$
\begin{aligned}
p_\mathcal{C} = \operatorname*{argmin}_{q \in \mathcal{P}_2(\mathbb{R}^d)} \quad & D(q\|p) \\
\text{subject to} \quad & \mathbb{P}_q(x \in \mathcal{C}) = 1.
\end{aligned}
\tag{3.3}
$$

*Proof.* For simplicity, we assume the existence of probability distributions. For $q \in \mathcal{P}_2(\mathbb{R}^d)$,

$$
\begin{aligned}
D(q\|p) &= \int_\mathcal{C} q(x) \log \frac{q(x)}{p(x)} \mathrm{d}x \\
&= \int_\mathcal{C} q(x) \log \left( \frac{q(x)}{p_\mathcal{C}(x)} \frac{Z_\mathcal{C}}{Z} \right) \mathrm{d}x \\
&= D(q\|p_\mathcal{C}) + \frac{Z_\mathcal{C}}{Z}
\end{aligned}
\tag{D.1}
$$

where $Z_\mathcal{C}$ satisfies

$$
\begin{aligned}
1 &= \int_{\mathbb{R}^d} p_\mathcal{C} \\
&= \frac{Z}{Z_\mathcal{C}} \int_\mathcal{C} p(x)\mathrm{d}x \\
&= \frac{Z}{Z_\mathcal{C}} \mathbb{P}_p(\mathcal{C}).
\end{aligned}
\tag{D.2}
$$

Therefore,

$$
D(q\|p) = D(q\|p_\mathcal{C}) + \mathbb{P}_p(\mathcal{C}).
\tag{D.3}
$$

This quantity is minimized for $q = p_\mathcal{C}$, and the minimal value is $\mathbb{P}_p(\mathcal{C})$. $\qquad\square$

### D.2 PROOF OF PROPOSITION 3.2

**Proposition 3.2** Strong duality is not attained for (3.3).

*Proof.* The Lagrangian of (3.3) is

$$
\begin{aligned}
L(q, \lambda) &:= D(q\|p) + \lambda \left( 1 - \mathbb{P}_q(x \in \mathcal{C}) \right) \\
&= D(q\|p) + \lambda \mathbb{E}_q[c(x)].
\end{aligned}
\tag{D.4}
$$

For all $\lambda \in \mathbb{R}$, the dual function is

$$
g(\lambda) := \min_{q \in \mathcal{P}_2(\mathbb{R}^d)} D(q\|p) + \lambda \mathbb{E}_q[c(x)]
\tag{D.5}
$$

and this minimum is attained by

$$
\begin{aligned}
p_\lambda(x) &= \frac{1}{Z_\lambda} \mathrm{e}^{-f(x) - \lambda c(x)} \\
&= \frac{Z}{Z_\lambda} p(x) \mathrm{e}^{-\lambda c(x)}.
\end{aligned}
\tag{D.6}
$$

The Lagrangian evaluated at $p_\lambda$ equals

$$
g(\lambda) = \log \frac{Z}{Z_\lambda}.
\tag{D.7}
$$

To compute $Z_\lambda$, we note that

$$1 = \int_{\mathbb{R}^d} p_\lambda$$
$$= \frac{Z}{Z_\lambda} \int_{\mathcal{C}} p(x) \mathrm{d}x + \frac{Z}{Z_\lambda} \int_{\bar{\mathcal{C}}} \mathrm{e}^{-\lambda c(x)} p(x) \mathrm{d}x \tag{D.8}$$

Let

$$\varepsilon(\lambda) := \int_{\bar{\mathcal{C}}} \mathrm{e}^{-\lambda c(x)} p(x) \mathrm{d}x. \tag{D.9}$$

Then,

$$1 = \frac{Z}{Z_\lambda} \left[ \mathbb{P}_p(\mathcal{C}) + \varepsilon(\lambda) \right] \tag{D.10}$$

By assumption, for all $\lambda \in \mathbb{R}^d$, $0 < \varepsilon(\lambda) < 1$. Furthermore, we obtain by combining (D.7) and (D.10), that

$$g(\lambda) = \log \frac{1}{\mathbb{P}_p(\mathcal{C}) + \varepsilon(\lambda)}. \tag{D.11}$$

This value is always strictly lower than its limit:

$$\forall \lambda, \; g(\lambda) < \log \frac{1}{\mathbb{P}_p(\mathcal{C})} = \lim_{\lambda \to +\infty} g(\lambda), \tag{D.12}$$

which is precisely the optimal value of Problem (3.3), attained by $q = p_\mathcal{C}$. Indeed,

$$D(p_\mathcal{C} \| p) = \int_{\mathcal{C}} \frac{Z}{Z_\mathcal{C}} p(x) \log \frac{Z}{Z_\mathcal{C}} \mathrm{d}x$$
$$= \mathbb{P}_p(\mathcal{C}) \frac{Z}{Z_\mathcal{C}} \log \frac{Z}{Z_\mathcal{C}}, \tag{D.13}$$

where $Z_\mathcal{C}$ satisfies

$$1 = \int_{\mathbb{R}^d} p_\mathcal{C}$$
$$= \frac{Z}{Z_\mathcal{C}} \int_{\mathcal{C}} p(x) \mathrm{d}x \tag{D.14}$$
$$= \frac{Z}{Z_\mathcal{C}} \mathbb{P}_p(\mathcal{C}).$$

It follows that

$$D(p_\mathcal{C} \| p) = \log \frac{1}{\mathbb{P}_p(\mathcal{C})}. \tag{D.15}$$

This value is found to be the minimizer of Problem (3.3) using Gibbs' inequality. $\square$

**Corollary 1** Penalty methods (2.5) cannot enforce $\mathbb{P}_q(\mathcal{C}) = 1$.

*Proof.* Penalty methods sample from $p_\lambda$, with finite $\lambda$. By the previous proof,

$$p_\lambda(x) \propto \exp\left(-f(x) - \lambda c(x)\right) \tag{D.16}$$

minimizes the $L(., \lambda)$. By definition of the dual function,

$$g(\lambda) = L(q_\lambda, \lambda). \tag{D.17}$$

For all densities $q \in \mathcal{P}_2(\mathbb{R}^d)$ satisfying the constraint $\mathbb{P}_q(\mathcal{C}) = 1$, the positive duality gap implies

$$L(q_\lambda, \lambda) < D(p_\mathcal{C} \| p) \le L(q, \lambda). \tag{D.18}$$

Therefore, $p_\lambda$ does not satisfy $\mathbb{P}_q(\mathcal{C}) = 1$. $\square$

## D.3 PROOF OF PROPOSITION 3.3

**Proposition 3.3** [Variable splitting] Problem (3.3) is equivalent to the following problem:

$$
\begin{aligned}
\underset{q\in\mathcal{P}_2(\mathbb{R}^d\times\mathbb{R}^d)}{\text{minimize}} \quad & D(q^x\|p) + \mathbb{E}_{q^z}\left[\chi_\mathcal{C}(z)\right] \\
\text{subject to} \quad & \mathbb{P}_q(x=z) = 1.
\end{aligned}
\tag{3.4}
$$

*Proof.* Given $q(x,z)$ the solution of Problem (3.4), the marginal $q^wx$ gives the solution of Problem (3.3). Given $q(x)$ the solution of Problem (3.3), the solution of Problem (3.4) can be obtained by defining $z$ as a copy of $x$. □

## D.4 PROOF OF PROPOSITION 3.4

**Proposition 3.4** [Strong duality] Strong duality holds and is attained for (P). In particular, the Lagrangian (3.5) admits a saddle point $(q_\star, \lambda_\star)$, and $q_\star$ is a solution of (P).

*Proof.* The expectation constraint is satisfied by the couple density of

$$
x \sim \mathcal{N}(0, I_d), \quad z|x \sim \mathcal{N}(x, I_d).
\tag{D.19}
$$

Hence, there exists a feasible point. Furthermore, the constraint function is surjective, as $\mathbb{E}_q[x-z] = v$ can be satisfied with $x \sim \mathcal{N}(0, I_d)$ and $z|x \sim \mathcal{N}(x+v, I_d)$. Therefore, Slater's constraint qualification condition is satisfied, and implies that strong duality holds and is attained. Relevant references can be found in Chamon et al. (2024). □

## D.5 PROOF OF PROPOSITION 4.1

**Lemma 1** [Integral formulation of the Lagrangian] The Lagrangian (3.5) of (P) is equal to

$$
L(q, \lambda) = D(q^x\|p) + \int\left[g(z) + \frac{\rho}{2}\|x+\mu-z\|^2\right]\mathrm{d}q^x(x)\mathrm{d}q(z|x) - \frac{\rho}{2}\|\mu\|^2,
\tag{D.20}
$$

with $\mu = (1/\rho)\times\lambda$.

**Proposition 4.1** The optimal distribution $q_\star^x$ takes the form

$$
q_\star^x(x) \propto \exp\left(-f(x) - \frac{\rho}{2}d_\mathcal{C}^2(x+\mu_\star)\right),
\tag{4.1}
$$

where $\mu_\star = (1/\rho)\times\lambda_\star$ and $d_\mathcal{C}(z) = \min_{y\in\mathcal{C}}\|y-z\|$ is the distance to the constraint set.

*Proof.* By Lemma 1 and by definition of the saddle point, $\mu_\star$ is such that $q_\star$ minimizes

$$
\begin{aligned}
L(q, \lambda_\star) &= D(q^x\|p) + \mathbb{E}_q\left[\frac{\rho}{2}\|Ax-z+\mu_\star\|^2 + \chi_\mathcal{C}(z),\right] - \frac{\rho}{2}\|\mu_\star\|^2 \\
&= D(q^x\|p) + \left[\frac{\rho}{2}\|Ax-z+\mu_\star\|^2 + \chi_\mathcal{C}(z)\right]\mathrm{d}q^x(x)\mathrm{d}q(z|x) - \frac{\rho}{2}\|\mu_\star\|^2.
\end{aligned}
\tag{D.21}
$$

Therefore, among the different conditionals $q(z|x)$, the optimal $q_\star(z|x)$ minimizes

$$
\int\left[\frac{\rho}{2}\|Ax-z+\mu_\star\|^2 + \chi_\mathcal{C}(z)\right]\mathrm{d}q_\star^x(x)\mathrm{d}q(z|x).
\tag{D.22}
$$

For each $x$, the function in brackets is minimized by

$$
z_\star(x) := P_\mathcal{C}(x+\mu_\star),
\tag{D.23}
$$

and hence, by integration against $q(z|x)$,

$$
\left[\frac{\rho}{2}\|Ax-z+\mu_\star\|^2 + \chi_\mathcal{C}(z)\right]\mathrm{d}q(z|x) \geq \frac{\rho}{2}\|Ax-z_\star(x)+\mu_\star\|^2 + \chi_\mathcal{C}(z_\star(x)).
\tag{D.24}
$$

This value is achieved by the deterministic conditional concentrated at $z_\star(x)$. Hence

$$
\forall x\in\mathbb{R}^d, \quad q_\star(z|x) = \delta_{z_\star(x)}(z).
\tag{D.25}
$$

By definition of $z_\star(x)$ and $g_\rho$, this implies that, for all $x$,

$$\int \left[ \frac{\rho}{2} \|Ax - z + \mu_\star\|^2 + \chi_{\mathcal{C}}(z) \right] \mathrm{d}q_\star(z|x) = \frac{\rho}{2} d_{\mathcal{C}}^2(x + \mu_\star) \tag{D.26}$$

Now replace this expression in the Lagrangian, seen as a function of $q^x$:

$$L(q, \mu_\star) = D(q^x \| p) + \int \frac{\rho}{2} d_{\mathcal{C}}^2(x + \mu_\star) \mathrm{d}q^x(x) - \frac{\rho}{2} \|\mu_\star\|^2$$
$$= \int \log q^x \mathrm{d}q^x + \mathbb{E}_{q^x}[f(x) + \frac{\rho}{2} d_{\mathcal{C}}^2(x + \mu_\star)] - \frac{\rho}{2} \|\mu_\star\|^2. \tag{D.27}$$

This functional is equal, up to a constant to the Kullback-Leibler divergence between $q^x$ and

$$q_\star(x) \propto \exp\left( -f(x) - \frac{\rho}{2} d_{\mathcal{C}}^2(x + \lambda_\star) \right), \tag{D.28}$$

and is hence minimized by the latter. $\qquad \square$

## D.6 PROOF OF PROPOSITION 4.2

**Proposition 4.2** [Recovery of the projection] Let $q_\rho$ denote the solution to (P). Then

$$q_\rho^x, q_\rho^z \xrightarrow[\rho \to +\infty]{} p_{\mathcal{C}} \quad \text{in distribution.} \tag{4.2}$$

*Proof of Proposition 4.2.* This result follows from Proposition 4.1. $\qquad \square$

## D.7 PROOF OF PROPOSITION 4.3

**Lemma 2** [Optimal value bound] Let $q_\star$ be a solution of Problem (P). Then

$$D(q_\star^x \| p) + \frac{\rho}{2} \mathbb{E}_{q_\star}[\|x - z\|^2] \leq D(p_{\mathcal{C}} \| p) \tag{D.29}$$

*Proof.* Let $q \in \mathcal{P}_2(\mathbb{R}^d \times \mathbb{R}^d)$ defined by $x \sim p_\star$ and $z|x \sim \delta_x$. Then $x \sim z \sim p_\star$, $q \in \mathcal{P}(\mathbb{R}^d \times \mathbb{R}^d)$, and $\mathbb{E}_q[x - z] = 0$, so $q$ is in the feasible set of Problem (P). This distribution provides the stated upper bound on the optimal value.

$\qquad \square$

**Proposition 4.3** [Relaxation error] The relaxed solution $q_\star$ satisfies the following conditions:

$$D(q_\star^x \| p) \leq D(p_{\mathcal{C}} \| p), \quad \mathbb{P}_{q_\star^z}(\mathcal{C}) = 1, \quad W_2^2(q_\star^x, q_\star^z) \leq \frac{1}{\rho} D(p_{\mathcal{C}} \| p). \tag{4.3}$$

*Proof.* The result follows from the previous lemma, noticing that

$$W_2^2(q_\star^x, q_\star^z) \leq \mathbb{E}_{q_\star}[\|x - z\|^2]. \tag{D.30}$$

$\qquad \square$

## D.8 PROOF OF PROPOSITION 4.4

**Proposition 4.4** [Consistency] In the case of a non-degenerate Gaussian potential $f$ and an affine constraint set $\mathcal{C}$, the solution of (P) is unbiased: $\mathbb{E}_{q_\star}[x] = \mathbb{E}_{q_\star}[z] = \mathbb{E}_{p_{\mathcal{C}}}[x]$.

*Proof.* Let $\mathcal{C} = \{x \in \mathbb{R}^d \mid Ax = b\}$ with $A \in \mathbb{R}^{(d-k) \times d}$ of full rank, $b \in \mathbb{R}^k$ and $k \geq 1$. We may parameterize $\mathcal{C} = v + W\mathbb{R}^k$, with $v := A^+ b \in \mathbb{R}^k$ and $W \in \mathbb{R}^{d \times k}$ a full column rank matrix gathering basis vectors of $\ker A$. Let $\bar{W} : \mathbb{R}^k \to \ker A$ denote the associated bijective operator.

We may assume that $p$ is a standard normal Gaussian, since operating a linear transform to $x$ preserves the problem with a different transformed affine constraint space. We thus assume that

$$f(x) = \frac{1}{2} x^\top x. \tag{D.31}$$

Let $y = \bar{W}^{-1}(x - v)$, and let $h : \mathcal{C} \to \mathbb{R}^n$ be a bounded function. Then,

$$
\begin{aligned}
\mathbb{E}[h(y)] &= \int_{\mathcal{C}} h(\bar{W}^{-1}(x - v)) \mathrm{d}p_{\mathcal{C}}(x) \\
&= \frac{Z}{Z_{\mathcal{C}}} \int_{\mathcal{C}} h(\bar{W}^{-1}(x - v)) \mathrm{e}^{-f(x)} \mathrm{d}x \\
&= \frac{Z}{Z_{\mathcal{C}}} |\bar{W}| \int_{\mathbb{R}^k} h(y) \mathrm{e}^{-f(v + \bar{W}y)} \mathrm{d}y \\
&= \frac{Z}{Z_{\mathcal{C}}} |\bar{W}| \int_{\mathbb{R}^k} h(y) \exp\left(-\frac{1}{2}(v + Wy)^{\top}(v + Wy)\right) \mathrm{d}y.
\end{aligned}
\tag{D.32}
$$

This shows that $y$ is a Gaussian vector, with quadratic potential

$$
\frac{1}{2}(v + Wy)^{\top}(v + Wy).
\tag{D.33}
$$

In particular, the expectation of $y$ maximizes this quadratic form:

$$
W^{\top}(W\mathbb{E}[y] + v) = 0.
\tag{D.34}
$$

Because $x = v + Wy$,

$$
\mathbb{E}_{p_{\mathcal{C}}}[x] = v + W\mathbb{E}[y].
\tag{D.35}
$$

Therefore,

$$
W^{\top}\mathbb{E}_{p_{\mathcal{C}}}[x] = 0,
\tag{D.36}
$$

which means exactly that $\mathbb{E}_{p_{\mathcal{C}}}[x] \in \mathcal{C}$ is the orthogonal projection of $\mathbb{E}_{p}[x] = 0$ onto $\mathcal{C}$. Recall that the projection onto $\mathcal{C}$ is equal to

$$
P_{\mathcal{C}}(x) = v + WW^{+}(x - v).
\tag{D.37}
$$

Second, note from Proposition 4.1 that $q_{\star}^{x}$ is Gaussian, as the projector is linear and

$$
d_{\mathcal{C}}^{2}(x) = \|P_{\mathcal{C}}(x) - x\|^2 = \|(WW^{+} - I_d)(x - v)\|^2.
\tag{D.38}
$$

The mean $m := \mathbb{E}_{q_{\star}^{x}}[x]$ under $q_{\star}^{x}$ maximizes the likelihood, and thus solves

$$
m + \rho(WW^{+} - I_d)^{\top}(WW^{+} - I_d)(m + \mu_{\star} - v) = 0,
\tag{D.39}
$$

which simplifies to

$$
m + \rho(I_d - WW^{+})(m + \mu_{\star} - v) = 0.
\tag{D.40}
$$

Left-multiplying by $W^{\top}$, we obtain that

$$
W^{\top}m = 0.
\tag{D.41}
$$

We obtain another equation by combining the equality constraint $\mathbb{E}_{q_{\star}}[x] = \mathbb{E}_{q_{\star}}[z]$, and the characterization of Proposition 4.1 implying $q_{\star}$-almost surely that $z = P_{\mathcal{C}}(x + \mu_{\star})$. By taking the expectation, we obtain

$$
m = \mathbb{E}_{q_{\star}}[x] = \mathbb{E}_{q_{\star}}[z] = v + WW^{+}(m + \mu_{\star} - v).
\tag{D.42}
$$

Substituting in (D.40), we obtain

$$
\nu + (\nu + \mu_{\star} - v) + v - \nu = 0,
\tag{D.43}
$$

implying that $m \in \mathcal{C}$. The two conditions $W^{\top}m = 0$ and $m \in \mathcal{C}$ characterize $m = \mathbb{E}_{q_{\star}}[x]$, and thus prove that

$$
\mathbb{E}_{q_{\star}}[x] = \mathbb{E}_{q_{\star}}[z] = \mathbb{E}_{p_{\mathcal{C}}}[x].
\tag{D.44}
$$

$\square$

### D.9    PROOF OF CONVERGENCE

We use the two following standard descent lemmas: one for the gradient step, and one for the projection step.

**Lemma 3** [Smooth gradient descent inequality] Let $f$ be a $\alpha$-convex and $\beta$-smooth function. Let $x \in \mathbb{R}^d$ and let $x_+ := x - \tau \nabla f(x)$. Then, for all $y \in \mathbb{R}^d$,

$$\frac{1}{2\tau}\|x - y\|^2 - \frac{1}{2\tau}\|x_+ - y\|^2 \geq f(x_+) - f(y) + \frac{1}{2\tau}(1 - \beta\tau)\|x_+ - x\|^2 + \frac{\alpha}{2}\|x - y\|^2 \quad \text{(D.45)}$$

**Lemma 4** [Projection inequality]  Let $\mathcal{C}$ be a convex set of $\mathbb{R}^k$. Let $z \in \mathbb{R}^k$. Then, for all $w \in \mathbb{R}^k$,

$$\|z - w\|^2 - \|P_{\mathcal{C}}(z) - w\|^2 \geq \|P_{\mathcal{C}}(z) - z\|^2. \quad \text{(D.46)}$$

We now study the effect of the alternated descent steps.

**Lemma 5** [Alternated gradient Lyapunov decrement] Let $x, \hat{x} \in \mathbb{R}^d$ and $z, \mu \in \mathbb{R}^k$. Define the iterations

$$\begin{aligned} x_+ &= x - \tau(Ax + \mu - z)) \\ z_+ &= z - \tau(z - A\hat{x} - \mu). \end{aligned} \quad \text{(D.47)}$$

Let  $\gamma = (\|A\|^2 + 1)$. Then for all $(y, w) \in \mathbb{R}^d \times \mathbb{R}^k$,

$$\begin{aligned} &\frac{1}{2\tau}\|(x - y, z - w)\|^2 - \frac{1}{2\tau}\|(x_+ - y, z_+ - w)\|^2 \geq \\ &\phi(\hat{x}, \hat{z}) - \phi(y, w) \\ &- \frac{1}{\tau}\|(x - x_+, z - z_+)\| \times \|x_+ - \hat{x}, z_+ - \hat{z}\| \\ &- \|A\| \times \|(x - \hat{x})\| \times \|w - \hat{z}\|. \\ &+ \frac{1}{2\tau}\|(x_+ - x, z_+ - z)\|^2 - \frac{\gamma}{2}\|(\hat{x} - x, \hat{z} - z)\|^2 \end{aligned} \quad \text{(D.48)}$$

*Proof.* The quadratic decrement factors as follows

$$\begin{aligned} &\frac{1}{2}\|(x - y, z - w)\|^2 - \frac{1}{2}\|(x_+ - y, z_+ - w)\|^2 \\ &= (x - x_+, z - z_+)^\top \left(\frac{1}{2}(x + x_+) - y, \frac{1}{2}(z + z_+) - w\right) \\ &= (x - x_+, z - z_+)^\top (x_+ - y, z_+ - w) \\ &+ (x - x_+, z - z_+)^\top \left(\frac{1}{2}(x - x_+), \frac{1}{2}(z - z_+)\right) \\ &= (x - x_+, z - z_+)^\top (\hat{x} - y, \hat{z} - w) + (x - x_+, z - z_+)^\top (x_+ - \hat{x}, z_+ - \hat{z}) + \frac{1}{2}\|(x_+ - x, z_+ - z)\|^2 \end{aligned} \quad \text{(D.49)}$$

Since $(x_+, z_+) = (x, z) - \tau \nabla \phi(x, z) + \tau(0, A(x - \hat{x}))$,

$$\nabla \phi(x, z)^\top (y - \hat{x}, w - \hat{z}) = \frac{1}{\tau}(x - x_+, z - z_+)^\top (y - \hat{x}, w - \hat{z}) + (x - \hat{x})^\top A^\top (w - \hat{z}). \quad \text{(D.50)}$$

Substituting in the quadratic decrement expression, we obtain

$$\begin{aligned} &\frac{1}{2\tau}\|(x - y, z - w)\|^2 - \frac{1}{2\tau}\|(x_+ - y, z_+ - w)\|^2 = \\ &- \nabla \phi(x, z)^\top (y - \hat{x}, w - \hat{z}) \\ &+ \frac{1}{\tau}(x - x_+, z - z_+)^\top (x_+ - \hat{x}, z_+ - \hat{z}) \\ &+ (x - \hat{x})^\top A^\top (w - \hat{z}) + \frac{1}{2\tau}\|(x_+ - x, z_+ - z)\|^2 \end{aligned} \quad \text{(D.51)}$$

By convexity of $\phi$,
$$\phi(y, w) \geq \phi(x, z) + \nabla\phi(x, z)^\top (y - x, w - z). \tag{D.52}$$

The function $\phi : (x, z) \mapsto \frac{1}{2}\|Ax - z + \mu\|^2$ is smooth. Letting $M := (A - I_d) \in \mathbb{R}^{d\times(k+d)}$, its Hessian is $M^\top M$, which has the same eigenvalues as $MM^\top = AA^\top + I_d$. Therefore, its smoothness constant is $\gamma = \|A\|^2 + 1$, with $\|A\|^2$ the largest singular value of $A$.

By smoothness,
$$\phi(\hat{x}, \hat{z}) \leq \phi(x, z) + \nabla\phi(x, z)^\top (\hat{x} - x, \hat{z} - z) + \frac{\gamma}{2}\|(\hat{x} - x, \hat{z} - z)\|^2. \tag{D.53}$$

Combining with the convexity inequality,
$$\phi(y, w) \geq \phi(\hat{x}, \hat{z}) + \nabla\phi(x, z)^\top (y - \hat{x}, w - \hat{z}) - \frac{\gamma}{2}\|(\hat{x} - x, \hat{z} - z)\|^2 \tag{D.54}$$

This implies the following quadratic decrement bound
$$\begin{aligned}
\frac{1}{2\tau}\|(x - y, z - w)\|^2 - \frac{1}{2\tau}\|(x_+ - y, z_+ - w)\|^2 \geq \\
\phi(\hat{x}, \hat{z}) - \phi(y, w) \\
+\frac{1}{\tau}(x - x_+, z - z_+)^\top (x_+ - \hat{x}, z_+ - \hat{z}) \\
+(x - \hat{x})^\top A^\top (w - \hat{z}) \\
+\frac{1}{2\tau}\|(x_+ - x, z_+ - z)\|^2 - \frac{\gamma}{2}\|(\hat{x} - x, \hat{z} - z)\|^2
\end{aligned} \tag{D.55}$$

Finally, by the Cauchy-Schwartz inequality
$$\frac{1}{\tau}(x - x_+, z - z_+)^\top (x_+ - \hat{x}, z_+ - \hat{z}) \geq -\frac{1}{\tau}\|(x - x_+, z - z_+)\| \times \|x_+ - \hat{x}, z_+ - \hat{z}\|. \tag{D.56}$$

and
$$(x - \hat{x})^\top A^\top (w - \hat{z}) \geq -\|A\| \times \|(x - \hat{x})\| \times \|w - \hat{z}\|. \tag{D.57}$$

$\square$

**Lemma 6** [Alternated proximal Lyapunov decrement] Let $x \in \mathbb{R}^d$ and $z, \mu \in \mathbb{R}^k$. Define the iterations
$$\begin{aligned}
x_+ &= \text{prox}_{\tau f}(x - \tau\rho(Ax + \mu - z)) \\
z_+ &= P_\mathcal{C}(z - \tau\rho(z - Ax_+ - \mu)).
\end{aligned} \tag{D.58}$$

Let $\gamma = \rho(\|A\|^2 + 1)$. Then for all $(y, w) \in \mathbb{R}^d \times \mathcal{C}$,
$$\begin{aligned}
\frac{1}{2\tau}\|(x - y, z - w)\|^2 - \frac{1}{2\tau}\|(x_+ - y, z_+ - w)\|^2 \geq \\
f(x_+) - f(y) + \frac{\rho}{2}\|Ax_+ + \mu - z_+\|^2 - \frac{\rho}{2}\|Ay + \mu - w\|^2 \\
+\frac{1}{2\tau}\|(x' - x, z' - z)\|^2 + \frac{1}{2\tau}(1 - \beta\tau)\|(x_+ - x', z_+ - z')\|^2 \\
+\frac{\alpha}{2}\|(x_+ - y, z_+ - w)\|^2 \\
-\frac{1}{\rho\tau}\|(x - x', z - z')\| \times \|x' - x_+, z' - z_+\| \\
-\rho\|A\| \times \|x - x_+\| \times D_\mathcal{C} \\
-\frac{\gamma}{2}\|(x_+ - x, z_+ - z)\|^2.
\end{aligned} \tag{D.59}$$

*Proof.* Let $x' := x - \tau\rho(Ax + \mu - z)$, and $z - \tau\rho(z - Ax_+ - \mu)$. Applying Lemma 5 to $x'$ and $z'$ with step size $\tau\rho$, and $\hat{x} = x_+$, we obtain for all $y \in \mathbb{R}^d$, $z \in \mathbb{R}^k$,

$$\frac{1}{2\tau}\|(x-y,z-w)\|^2 - \frac{1}{2\tau}\|(x'-y,z'-w)\|^2 \geq$$
$$\frac{1}{2}\rho\|Ax_+ + \mu - z_+\|^2 - \frac{1}{2}\rho\|Ay + \mu - w\|^2$$
$$+\frac{1}{2\tau}\|(x'-x,z'-z)\|^2$$
$$-\frac{1}{\rho\tau}\|(x-x',z-z')\| \times \|x'-x_+,z'-z_+\|$$
$$-\rho\|A\| \times \|x-x_+\| \times \|w-z_+\|$$
$$-\frac{\gamma}{2}\|(x_+-x,z_+-z)\|^2,$$

(D.60)

with $\gamma = \rho(\|A\|^2+1)$.

We bound

$$\|w-z_+\| \leq D_{\mathcal{C}},$$

(D.61)

and

$$\frac{1}{\rho\tau}\|(x-x',z-z')\| = \|(x-z+\mu, z-x\mu)\| \leq M.$$

(D.62)

Furthermore, applying and summing the descent inequalities for $f$ and $P_{\mathcal{C}}$ at $x'$ and $z'$,

$$\frac{1}{2\tau}\|x'-y\|^2 - \frac{1}{2\tau}\|x_+-y\|^2$$
$$+\frac{1}{2\tau}\|z'-w\|^2 - \frac{1}{2\tau}\|z_+-w\|^2 \geq$$
$$f(x_+) - f(y) + \frac{1}{2\tau}(1-\beta\tau)\|x_+-x'\|^2 + \frac{\alpha}{2}\|x_+-y\|^2$$
$$+\frac{1}{2\tau}\|z_+-z'\|^2$$

(D.63)

The result is obtained by adding these inequalities.

$\square$

## D.10 PROOF OF PROPOSITION 4.5

**Lemma 7** [Primal-dual Lyapunov decrement] Let $x \in \mathbb{R}^d$ and $z, \mu \in \mathbb{R}^k$. Define the iterations

$$x_+ = \text{prox}_{\tau f}(x - \tau\rho(Ax + \mu - z))$$
$$z_+ = P_{\mathcal{C}}(z - \tau\rho(z - Ax_+ - \mu).$$
$$\mu_+ = \mu + \tau(x_+ - z_+).$$

(D.64)

Let $\gamma = \rho(\|A\|^2+1)$. Then for all $(y,w,\nu) \in \mathbb{R}^d \times \mathcal{C} \times \mathbb{R}^k$, provided $4\tau(\beta + 2\rho(\|A\|^2+1)) \leq 1$,

$$\frac{1}{2\tau}\|(x-y,z-w)\|^2 - \frac{1}{2\tau}\|(x_+-y,z_+-w)\|^2 + \frac{1}{2\tau}\|\lambda-\nu\|^2 - \frac{1}{2\tau}\|\lambda_+-\nu\|^2 \geq$$
$$f(x_+) - f(y) + \frac{\rho}{2}\|Ax_+-z_+\|^2 - \frac{\rho}{2}\|Ay-w\|^2$$
$$+\nu^\top(Ax_+-z_+) - \lambda^\top(Ay-w)$$
$$+\frac{\alpha}{2}\|(x_+-y,z_+-w)\|^2$$
$$-4\tau\|A\|^2 D_{\mathcal{C}}^2 - 2M\tau^2 - \frac{\rho\tau}{2}\|Ax_+-z_+\|^2.$$

(D.65)

*Proof.* By Lemma 6,

$$
\begin{aligned}
\frac{1}{2\tau}\|(x-y, z-w)\|^2 - &\frac{1}{2\tau}\|(x_+ - y, z_+ - w)\|^2 \geq \\
& f(x_+) - f(y) + \frac{\rho}{2}\|Ax_+ + \mu - z_+\|^2 - \frac{\rho}{2}\|Ay + \mu - w\|^2 \\
& + \frac{1}{2\tau}\|(x'-x, z'-z)\|^2 + \frac{1}{2\tau}(1-\beta\tau)\|(x_+ - x', z_+ - z')\|^2 \\
& + \frac{\alpha}{2}\|(x_+ - y, z_+ - w)\|^2 \\
& - \frac{\rho}{\tau}\|(x-x', z-z')\| \times \|x' - x_+, z' - z_+\| \\
& - \rho\|A\| \times \|x - x_+\| \times \|w - z_+\| \\
& - \frac{\gamma}{2}\|(x_+ - x, z_+ - z)\|^2.
\end{aligned}
\tag{D.66}
$$

We are going to absorb the negative terms of the right-hand side into the positive quadratic terms, up to terms of order $\tau$.

• First decompose

$$
\|(x_+ - x, z_+ - z)\|^2 \leq 2\|(x_+ - x', z_+ - z')\|^2 + 2\|(x' - x, z' - z)\|^2,
\tag{D.67}
$$

so that

$$
-\frac{\gamma}{2}\|(x_+ - x, z_+ - z)\|^2 \geq -\gamma\|(x_+ - x', z_+ - z')\|^2 - \gamma\|(x' - x, z' - z)\|^2.
\tag{D.68}
$$

By absorbing these contributions, the remaining quadratic terms are

$$
\frac{1}{2\tau}(1-2\gamma\tau)\|(x'-x, z'-z)\|^2 + \frac{1}{2\tau}(1-(\beta+2\gamma)\tau)\|(x_+ - x', z_+ - z')\|^2
\tag{D.69}
$$

• By combining the Cauchy-Schwartz inequality and Young's inequality, for all $\sigma > 0$,

$$
\rho\|A\| \times \|x - x_+\| \times \|w - z_+\| \geq -\frac{\sigma}{2}\|x - x_+\|^2 - \frac{1}{2\sigma}\rho^2\|A\|^2 D_{\mathcal{C}}^2.
\tag{D.70}
$$

In order to absorb these terms in the quadratic terms, we choose $\sigma = 1/(4\tau)$ and obtain

$$
\begin{aligned}
\rho\|A\| \times \|x - x_+\| \times \|w - z_+\| &\geq -\frac{1}{8\tau}\|x - x_+\|^2 - 4\tau\rho^2\|A\|^2 D_{\mathcal{C}}^2 \\
&\geq -\frac{1}{8\tau}\|(x - x_+, z - z_+)\|^2 - 4\tau\rho^2\|A\|^2 D_{\mathcal{C}}^2. \\
&\geq -\frac{1}{4\tau}\|(x - x', z - z')\|^2 - \frac{1}{4\tau}\|(x' - x_+, z' - z_+)\|^2 - 4\tau\rho^2\|A\|^2 D_{\mathcal{C}}^2.
\end{aligned}
\tag{D.71}
$$

After absorbing the first term, the remaining quadratic terms are

$$
\frac{1}{2\tau}(1/2 - 2\gamma\tau)\|(x'-x, z'-z)\|^2 + \frac{1}{2\tau}(1/2 - (\beta+2\gamma)\tau)\|(x_+ - x', z_+ - z')\|^2
\tag{D.72}
$$

By a similar argument,

$$
\begin{aligned}
\frac{\rho}{\tau}\|(x - x', z - z')\| \times \|(x' - x_+, z' - z_+)\| &\geq -\frac{1}{8\tau}\|(x' - x_+, z' - z_+)\|^2 - 2\rho^2 M^2 \tau \\
&\geq -\frac{1}{8\tau}\|(x - x', z - z')\|^2 - \frac{1}{8\tau}\|(x' - x_+, z' - z_+)\|^2 - 2\rho^2 M^2 \tau.
\end{aligned}
\tag{D.73}
$$

After absorbing the first two terms, the remaining quadratic terms are

$$
\frac{1}{2\tau}(1/4 - 2\gamma\tau)\|(x'-x, z'-z)\|^2 + \frac{1}{2\tau}(1/4 - (\beta+2\gamma)\tau)\|(x_+ - x', z_+ - z')\|^2
\tag{D.74}
$$

Finally, provided $4(2\gamma + \beta)\tau \leq 1$,

$$
\begin{aligned}
\frac{1}{2\tau}\|(x - y, z - w)\|^2 &- \frac{1}{2\tau}\|(x_+ - y, z_+ - w)\|^2 \geq \\
&f(x_+) - f(y) + \frac{\rho}{2}\|Ax_+ + \mu - z_+\|^2 - \frac{\rho}{2}\|Ay + \mu - w\|^2 \\
&+ \frac{\alpha}{2}\|(x_+ - y, z_+ - w)\|^2 \\
&- 4\tau\rho^2\|A\|^2 D_{\mathcal{C}}^2 - 2\rho^2 M^2\tau.
\end{aligned}
\tag{D.75}
$$

Now note that

$$
\|Ax_+ + \mu - z_+\|^2 - \|Ay + \mu - w\|^2 = \|Ax_+ - z_+\|^2 - \|Ay - w\|^2 + \mu^\top(Ax_+ - z_+ - (Ay - w)),
\tag{D.76}
$$

where $\rho\mu = \lambda$.

By the dual iteration $\lambda_+ = \lambda + \tau(Ax_+ - z_+)$,

$$
\begin{aligned}
\frac{1}{2\tau}\|\lambda_+ - \nu\|^2 &= \frac{1}{2\tau}\|\lambda - \nu\|^2 + \frac{\tau}{2}\|Ax_+ - z_+\|^2 + \frac{1}{\tau}(\lambda - \nu)^\top(\lambda_+ - \lambda) \\
&= \frac{1}{2\tau}\|\lambda - \nu\|^2 + \frac{\tau}{2}\|Ax_+ - z_+\|^2 + (\lambda - \nu)^\top(Ax_+ - z_+),
\end{aligned}
\tag{D.77}
$$

implying

$$
\frac{1}{2\tau}\|\lambda - \nu\|^2 - \frac{1}{2\tau}\|\lambda_+ - \nu\|^2 = -\frac{\tau}{2}\|Ax_+ - z_+\|^2 + (\nu - \lambda)^\top(Ax_+ - z_+).
\tag{D.78}
$$

Gathering these equations,

$$
\begin{aligned}
\frac{1}{2\tau}\|(x - y, z - w)\|^2 &- \frac{1}{2\tau}\|(x_+ - y, z_+ - w)\|^2 + \frac{1}{2\tau}\|\lambda - \nu\|^2 - \frac{1}{2\tau}\|\lambda_+ - \nu\|^2 + \geq \\
&f(x_+) - f(y) + \frac{\rho}{2}\|Ax_+ - z_+\|^2 - \frac{\rho}{2}\|Ay - w\|^2 \\
&+ \nu^\top(Ax_+ - z_+) - \rho\lambda^\top(Ay - w) \\
&+ \frac{\alpha}{2}\|(x_+ - y, z_+ - w)\|^2 \\
&- 4\tau\|A\|^2 D_{\mathcal{C}}^2 - 2\rho^2 M^2\tau - \frac{\rho\tau}{2}\|Ax_+ - z_+\|^2.
\end{aligned}
\tag{D.79}
$$

$\square$

**Proposition 4.5** [Primal-dual Lyapunov decrement] Let $q_+$ be the distribution after iterations (3.6) with step size $\tau$. Then, provided $4\tau(\beta + 4\rho) \leq 1$, for any reference pair $(p, \nu)$:

$$
V(q, \lambda; p, \nu) - \mathbb{E}[V(q_+, \lambda_+; p, \nu)|\lambda] + \frac{\alpha}{2}W_2^2(q, q_+) \geq \tau(L(q_+, \nu) - L(q_\star, \lambda)) + C\tau^2. \tag{4.5}
$$

*Proof.* Let $(x_+, z_+) \sim q_+$. Note that $x_+ := \bar{x}_+ + \sqrt{2\tau}w$, where $(\bar{x}_+, z_+)$ is defined in the previous lemma, and $w \sim \mathcal{N}(0, I_d)$ is an independent Gaussian noise. We let $\bar{q}_+$ be the distribution of $(\bar{x}_+, z_+)$.

The previous lemma is valid for all $(x, z), (y, w)$. Choose $y(x)$ to be the optimal transport plan between $q^x$ and $p^x$, and $w(z)$ to be the optimal transport plan between $q^z$ and $p^z$. Then

$$
\mathbb{E}_q[\|x - y\|^2] = W_2^2(q^x, p^x) \tag{D.80a}
$$

$$
\mathbb{E}_q[\|z - w\|^2] = W_2^2(q^z, p^z). \tag{D.80b}
$$

Furthermore, by definition of the Wasserstein distance,

$$
\mathbb{E}_q[\|\bar{x}_+ - y\|^2] \geq W_2^2(q_+^x, p^x) \tag{D.81a}
$$

$$
\mathbb{E}_q[\|z_+ - w\|^2] \geq W_2^2(q_+^z, p^z), \tag{D.81b}
$$

and
$$W_2^2(q^x, \bar{q}_+^x) + W_2^2(q^z, q_+^z) \leq \mathbb{E}_q\left[\|(\bar{x}_+ - y, z_+ - w)\|^2\right]. \tag{D.82}$$

By integration of Lemma 7 against $q$,

$$\frac{1}{2\tau}W_2^2(q^x, p^x) + \frac{1}{2\tau}W_2^2(q^z, p^z) - \frac{1}{2\tau}W_2^2(q^x, \bar{q}_+^x) - \frac{1}{2\tau}W_2^2(q^z, q_+^z)$$
$$+\frac{1}{2\tau}\mathbb{E}_q\left[\|\lambda - \nu\|^2\right] - \frac{1}{2\tau}\mathbb{E}_q\left[\|\lambda_+ - \nu\|^2\right]$$
$$+\frac{\alpha}{2}\mathbb{E}_q[\|\bar{x}_+ - x\|^2] \geq \tag{D.83}$$
$$\mathbb{E}_q[f(\bar{x}_+) + \frac{\rho}{2}\|A\bar{x}_+ - z_+\|^2] - \mathbb{E}_p[f(y) + \frac{\rho}{2}\|Ay - w\|^2]$$
$$+\rho\nu^\top\mathbb{E}_q\left[A\bar{x}_+ - z_+\right] - \rho\nu^\top\mathbb{E}_p\left[Ay - w\right]$$
$$-4\tau\|A\|^2\|w - z_+\|^2 - 2\rho^2 M^2\tau - \frac{\rho\tau}{2}\|A\bar{x}_+ - z_+\|^2.$$

Note that
$$\|A\bar{x}_+ - z_+\|^2 = \|Ax_+ - z_+ - \sqrt{2\tau}Aw\|^2$$
$$= \|Ax_+ - z_+\|^2 + 2\tau\|Aw\|^2 - \sqrt{2\tau}w^\top(Ax_+ - z_+), \tag{D.84}$$

whose expectation is
$$\mathbb{E}_q\left[\|A\bar{x}_+ - z_+\|^2\right] = \mathbb{E}_q\left[\|Ax_+ - z_+\|^2\right] + 2\tau\|A\|_2^2. \tag{D.85}$$

The other quadratic terms in $\bar{x}_+$ are handled similarly. Expectations of linear terms in $\bar{x}_+$ are not affected by the Gaussian noise. We thus obtain

$$\frac{1}{2\tau}W_2^2(q^x, p^x) + \frac{1}{2\tau}W_2^2(q^z, p^z) - \frac{1}{2\tau}W_2^2(q^x, \bar{q}_+^x) - \frac{1}{2\tau}W_2^2(q^z, q_+^z)$$
$$+\frac{1}{2\tau}\mathbb{E}_q\left[\|\lambda - \nu\|^2\right] - \frac{1}{2\tau}\mathbb{E}_q\left[\|\lambda_+ - \nu\|^2\right]$$
$$+\frac{\alpha}{2}W_2^2(q_+, q) \geq \tag{D.86}$$
$$\mathbb{E}_q[f(x_+) + \frac{\rho}{2}\|Ax_+ - z_+\|^2] - \mathbb{E}_p[f(y) + \frac{\rho}{2}\|Ay - w\|^2]$$
$$+\rho\nu^\top\mathbb{E}_q\left[Ax_+ - z_+\right] - \rho\nu^\top\mathbb{E}_p\left[Ay - w\right]$$
$$-4\tau\|A\|^2\|w - z_+\|^2 - 2\rho^2 M^2\tau - \frac{\rho\tau}{2}\|Ax_+ - z_+\|^2 + (2\rho\|A\| + d\alpha)\tau.$$

In order to control the value of $\mathbb{E}_{q_+}[f]$, we use the $\beta$-smoothness assumption and apply Lemma 3 of Durmus et al. (2019):
$$\mathbb{E}_{q_+}[f(x)] - \mathbb{E}_{\bar{q}_+}[f(x)] \leq \beta d\tau \tag{D.87}$$

We finally handle the entropy terms. For $\varphi \in \mathcal{P}_2(\mathbb{R}^d)$, let
$$H(\varphi) := \int \varphi \log \varphi. \tag{D.88}$$

Since $x_+ = \bar{x}_+ + \sqrt{2\tau}w$, Lemma 5 of Durmus et al. (2019) applies and yields
$$\frac{1}{2\tau}W_2^2(\bar{q}^x, p^x) - \frac{1}{2\tau}W_2^2(q_+^x, p^x) \geq H(q_+^x) - H(p^x). \tag{D.89}$$

Gathering the inequalities above, we obtain

$$\frac{1}{2\tau}W_2^2(q^x, p^x) + \frac{1}{2\tau}W_2^2(q^z, p^z) - \frac{1}{2\tau}W_2^2(q^x, \bar{q}_+^x) - \frac{1}{2\tau}W_2^2(q^z, q_+^z)$$
$$+\frac{1}{2\tau}\mathbb{E}_q\left[\|\lambda - \nu\|^2\right] - \frac{1}{2\tau}\mathbb{E}_q\left[\|\lambda_+ - \nu\|^2\right]$$
$$+\frac{\alpha}{2}W_2^2(q_+, q) \geq \tag{D.90}$$
$$L(q_+, \nu) - L(p, \mu)$$
$$-C\tau.$$

with

$$C = -4\tau \|A\|^2 \|w - z_+\|^2 - 2\rho^2 M^2 \tau + \frac{\rho}{2}\|Ax_+ - z_+\|^2 + (2\rho\|A\| + d\alpha) + \beta d. \qquad \text{(D.91)}$$

$\square$

### D.11   PROOF OF PROPOSITION D.1

**Lemma 8** [Saddle gap bound]  Let $(q_\star, \lambda_\star)$ be the saddle point of Problem (P). Then, for all $(q, \lambda) \in \mathcal{P}_2(\mathbb{R}^d) \times \mathbb{R}^k$,

$$L(q, \lambda_\star) - L(q_\star, \lambda) \geq D(q^x\|q_\star^x). \qquad \text{(D.92)}$$

*Proof.* By definition of $(q_\star, \lambda_\star)$, $L(q_\star, \lambda) = L(q_\star, \lambda_\star)$. Furthermore, by Lemma 1,

$$
\begin{aligned}
L(q, \lambda_\star) - L(q_\star, \lambda_\star) =& D(q^x\|p) + \int \left[\frac{\rho}{2}\|Ax - z + \mu_\star\|^2 + \chi_{\mathcal{C}}(z)\right] \mathrm{d}q^x(x)\mathrm{d}q(z|x) \\
& - D(q_\star^x\|p) + \int \left[\frac{\rho}{2}\|Ax - z + \mu_\star\|^2 + \chi_{\mathcal{C}}(z)\right] \mathrm{d}q_\star^x(x)\mathrm{d}q_\star(z|x) \\
=& D(q^x\|p) - D(q_\star^x\|p) \\
& + \int \left[\frac{\rho}{2}\|Ax - z + \mu_\star\|^2 + \chi_{\mathcal{C}}(z)\right] \mathrm{d}q^x(x) \left(\mathrm{d}q(z|x) - \mathrm{d}q_\star(z|x)\right) \\
& + \int \left[\frac{\rho}{2}\|Ax - z + \mu_\star\|^2 + \chi_{\mathcal{C}}(z)\right] \left(\mathrm{d}q^x(x) - \mathrm{d}q_\star^x(x)\right) \mathrm{d}q_\star(z|x).
\end{aligned}
\qquad \text{(D.93)}
$$

From the characterization of $q_\star$ in Proposition 4.1, $q_\star(z|x) = \delta_{z_\star(x)}(z)$, and

$$\int \left[\frac{\rho}{2}\|Ax - z + \mu_\star\|^2 + \chi_{\mathcal{C}}(z)\right] \mathrm{d}q^x(x) \left(\mathrm{d}q(z|x) - \mathrm{d}q_\star(z|x)\right) \geq 0. \qquad \text{(D.94)}$$

Therefore,

$$L(q, \lambda_\star) - L(q_\star, \lambda_\star) \geq D(q^x\|p) - D(q_\star^x\|p) + \int \frac{\rho}{2} d_{\mathcal{C}}^2(x + \mu_\star) \left(\mathrm{d}q^x(x) - \mathrm{d}q_\star^x(x)\right), \qquad \text{(D.95)}$$

that is

$$
\begin{aligned}
L(q, \lambda_\star) - L(q_\star, \lambda_\star) \geq& \int \log q^x \mathrm{d}q^x + \mathbb{E}_q[f(x) + \frac{\rho}{2}d_{\mathcal{C}}^2(x + \mu_\star)] \\
& - \int \log q_\star^x \mathrm{d}q_\star^x - \mathbb{E}_{q_\star}[f(x) + \frac{\rho}{2}d_{\mathcal{C}}^2(x + \mu_\star)]
\end{aligned}
\qquad \text{(D.96)}
$$

$$L(q, \lambda_\star) - L(q_\star, \lambda_\star) \geq D(q^x\|q_\star^x) - D(q_\star^x\|q_\star^x) = D(q^x\|q_\star^x).$$

$\square$

**Proposition D.1** [Mixing rate]  Then, there exists $C > 0$ such that

$$\sum_{t=0}^{T-1} \tau_t D(q_{t+1}^x\|q_\star^x) + \mathbb{E}[V(q_T, \lambda_T)] \leq C \sum_{t=0}^{T-1} \tau_t^2 + V(q_0, \lambda_0). \qquad \text{(D.97)}$$

*Proof.* Let $\bar{q}_t$ be the probability measure of $(x_t, z_t)$ conditionally on $\lambda_0, \dots, \lambda_t$, and note that $(\bar{q}_{t+1}, \lambda_{t+1})$ is obtained from $\bar{q}_t, \lambda_t$ by applying updates (3.6). We apply Proposition 4.5 to $\bar{q}_t$ and $(p, \nu) = (q_\star, \lambda_\star)$, and apply Lemma 8 to bound

$$L(\bar{q}_t, \lambda_\star) - L(q_\star, \lambda_t) \geq D(\bar{q}_t^x\|q_\star^x). \qquad \text{(D.98)}$$

We obtain

$$
\begin{aligned}
& \frac{1}{2\tau}W_2^2(\bar{q}_t^x, q_\star^x) + \frac{1}{2\tau}W_2^2(\bar{q}_t^z, q_\star^z) - \frac{1}{2\tau}W_2^2(\bar{q}_t^x, \bar{q}_{t+1}^x) - \frac{1}{2\tau}W_2^2(\bar{q}_t^z, \bar{q}_{t+1}^z) \\
& + \frac{1}{2\tau}\mathbb{E}_q\left[\|\lambda_t - \lambda_\star\|^2\right] - \frac{1}{2\tau}\mathbb{E}_q\left[\|\lambda_{t+1} - \lambda_\star\|^2\right] \\
& + \frac{\alpha}{2}W_2^2(\bar{q}_{t+1}, \bar{q}_t) \geq \\
& D(\bar{q}_{t+1}^x\|q_\star^x) - C\tau.
\end{aligned}
\qquad \text{(D.99)}
$$

Summing these inequalities for $1 \leq t \leq T-1$,

$$
\sum_{t=0}^{T-1} \tau_t D(\bar{q}_{t+1}^x \| q_\star^x) + \frac{1}{2} W_2^2(\bar{q}_T^x, q_\star^x) + \frac{1}{2} W_2^2(\bar{q}_T^z, q_\star^z) + \frac{1}{2} \|\lambda_T - \lambda_\star\|^2 \leq
$$
$$
\frac{1}{2} W_2^2(q_0^x, q_\star^x) + \frac{1}{2} W_2^2(q_0^z, q_\star^z) + \frac{1}{2} \|\lambda_0 - \lambda_\star\|^2 + C \sum_{t=0}^{T-1} \tau_t^2. \tag{D.100}
$$

We finish the proof by taking the expectation over $\Lambda_T := \lambda_0, \ldots, \lambda_T$. To compute the average of the mutual information, note that

$$
\mathbb{E}\left[\mathbb{E}[f(x_t) | \Lambda_T]\right] = \mathbb{E}[f(x_t)], \tag{D.101}
$$

and, by the entropy

$$
\mathbb{E}[H(\bar{q}_t^x)] = \mathbb{E}[H(q(x_t | \Lambda_t)] 
$$
$$
\geq H(q_t^x). \tag{D.102}
$$

We finally obtain

$$
\sum_{t=0}^{T-1} \tau_t D(q_{t+1}^x \| q_\star^x) + \frac{1}{2} \mathbb{E}[W_2^2(q_T^x, q_\star^x)] + \frac{1}{2} \mathbb{E}[W_2^2(q_T^z, q_\star^z)] + \frac{1}{2} \mathbb{E}[\|\lambda_T - \lambda_\star\|^2] \leq
$$
$$
\frac{1}{2} W_2^2(q_0^x, q_\star^x) + \frac{1}{2} W_2^2(q_0^z, q_\star^z) + \frac{1}{2} \|\lambda_0 - \lambda_\star\|^2 + C \sum_{t=0}^{T-1} \tau_t^2. \tag{D.103}
$$

**Corollary 2** [Mixing rate] Let $(x_t)$ be generated by Algorithm 1 with step size $\tau_t := 1/\sqrt{t+1}$. Let $\bar{q}_t$ denote the distribution of the time-averaged iterate $\bar{x}_t := \frac{1}{t} \sum_{s=0}^{t-1} x_s$. Then,

$$
D(\bar{q}_t \| q_\star^x) \leq \mathcal{O}\left(\frac{\ln t}{\sqrt{t}}\right). \tag{4.6}
$$

*Proof of Corollary 2.* By convexity of the Kullback-Leibler divergence (Cover, 1999),

$$
D(\bar{q}_t^x \| q_\star^x) \leq \frac{1}{t} \sum_{s=0}^{t-1} D(q_s^x \| q_\star^x) \tag{D.104}
$$

Applying Proposition D.1 to $\tau_s = 1/\sqrt{t}$,

$$
\frac{1}{\sqrt{t}} \sum_{s=0}^{t-1} D(q_{s+1}^x \| q_\star^x) + \mathbb{E}[V(q_t, \lambda_T)] \leq
$$
$$
C \sum_{s=0}^{t-1} \frac{1}{s} + V(q_0, \lambda_0). \tag{D.105}
$$

This implies

$$
\frac{1}{t} \sum_{s=0}^{t-1} D(q_{s+1}^x \| q_\star^x) \leq \frac{C}{\sqrt{t}} \sum_{s=0}^{t-1} \frac{1}{s} + \frac{1}{\sqrt{t}} V(q_0, \lambda_0), \tag{D.106}
$$

and

$$
\sum_{s=0}^{t-1} \frac{1}{s} \underset{t \to +\infty}{\sim} \log t. \tag{D.107}
$$

Therefore,

$$
D(\bar{q}_t^x \| q_\star^x) \underset{t \to +\infty}{=} \mathcal{O}\left(\log t / \sqrt{t}\right). \tag{D.108}
$$

$\square$

$\square$

**Usage of Large Language Models** We used large language models at the sentence level to correct English writing and avoid word repetition.

