# OpenReview forum: "Strictly Constrained Generative Modeling via Split Augmented Langevin Sampling"
_ICLR.cc/2026/Conference — ICLR 2026 Poster_

### Official Review · Reviewer_1KBT · 2025-10-30

**Soundness:** 3
**Presentation:** 2
**Contribution:** 3
**Rating:** 4
**Confidence:** 2

**Summary:**

This paper tackles the problem of sampling from generative models while strictly satisfying physical constraints like conservation laws. Existing approaches either fail to enforce constraints exactly (penalty methods) or become trapped in limited regions under non-convex constraints (projection methods). The authors propose Split Augmented Langevin (SAL), which uses variable splitting and augmented Lagrangian dynamics to progressively enforce constraints while preserving exploration capability. The method operates in a training-free manner on pre-trained diffusion models and provides convergence guarantees through duality analysis. Experiments on energy-preserving field generation, data assimilation for the Burgers equation, and optimal control problems demonstrate that SAL achieves both strict constraint satisfaction and accurate conditional sampling, outperforming existing baselines in maintaining physical plausibility and forecast accuracy.

**Strengths:**

**Principled framework with theoretical guarantees:** The variational formulation and duality analysis provide strong theoretical foundations, proving that the relaxed problem recovers the target distribution as coupling strength increases.

**Training-free and modular:** The method works as a drop-in replacement for standard Langevin steps in pre-trained diffusion models, requiring no retraining or additional data.

**Strong empirical validation:** Experiments on diverse physical systems demonstrate practical effectiveness, showing improved constraint satisfaction and forecast accuracy compared to existing baselines.

**Weaknesses:**

**Limited Complexity of Constraints.** While the experiments effectively demonstrate the method's core capabilities, the constraint types considered are relatively simple. The energy conservation constraint in Section 5.1 is a quadratic sphere, and the mass conservation in Section 5.2 is linear. Real-world physical systems often involve more complex constraints, such as coupled higher-order nonlinear conservation laws (e.g., multi-component energy functionals with nonlinear operators), or systems governed by multiple interacting PDEs with intricate coupling structures. It remains unclear how the projection step and the coupling parameter would perform when the constraint manifold has significantly more complex geometry or when the projection itself becomes computationally expensive. Evaluating SAL on such challenging constraints would strengthen confidence in its applicability to complex physical modeling tasks.

**Gap between Theoretical Analysis and Actual Implementation.** While the paper presents a practical and effective algorithm, there is a notable gap between the theoretical analysis and the actual implementation. The convergence analysis in Section 4.3 focuses on the optimization problem (Eq 4.4) in the space of probability measures, establishing strong duality (Proposition 5) and asymptotic recovery (Proposition 6). However, the actual algorithm (Eq 4.5-4.6) is a stochastic sampling procedure operating in sample space with Langevin noise. The missing piece is a rigorous convergence analysis of the proposed iterative scheme (4.6) itself, which would require establishing: (1) Non-asymptotic convergence rates in Wasserstein distance, (2) Finite-sample guarantees for fixed $\rho$ values, (3) Guidance on hyperparameter selection ($\rho, \tau, \eta$) based on convergence theory. The current theoretical results only guarantee that as $\rho \to \infty$, the solution to (4.4) approaches $p_C$, but do not characterize how the finite-step stochastic algorithm (4.6) approximates this solution. This is acknowledged by the authors as a limitation, but addressing it would significantly strengthen the contribution. Despite this gap, the empirical results convincingly demonstrate the method's effectiveness.

**Questions:**

While the theoretical contributions (Propositions 3-6) are rigorous, the connection between the mathematical advantages and the observed performance improvements in Section 5 is not clearly articulated. Specifically, it remains unclear how the duality gap in Proposition 3 directly explains the constraint violations in Figure 3, or how Proposition 5's attained duality resolves both the exploration bias (Figure 2) and projection artifacts (Figure 4) simultaneously. I recommend adding explicit statements or a summary that maps each theoretical result to the corresponding experimental observations, making the paper more accessible to readers outside convex optimization. A comparative table contrasting penalty methods, projected Langevin, and SAL across key mathematical properties (duality, constraint satisfaction, exploration) would significantly improve clarity.

---

> ### Author Response · Authors · 2025-11-15
> **Response to  Reviewer 1KBT, answer to weaknesses**
>
> We thank the reviewer for the detailed and constructive feedback. We found the comments very helpful and used them to clarify theoretical aspects and strengthen the connection between our analysis and experiments. Below we address each point in turn.
>
> ## Answers to weaknesses
>
>
> `Limited Complexity of Constraints. `
>
> We appreciate the reviewer’s comment and fully agree that the complexity of the projection operator was not sufficiently highlighted in the main text.
> As detailed in our general clarification above, Experiment 5.2 already involves a non-analytical high-dimensional projection, taking the form of a non-convex constrained optimization problem, implemented via a numerical iterative algorithm.
> We now emphasize this point more clearly in the revised version and summarize the associated runtime study from Appendix A.8 in the main text.
>
> More precisely, in the Burgers equation experiment, states must satisfy mass and energy conservation:
>
> $$
> \mathcal{C} = \{z \in \mathbb{R}^d \mid 1^{\top} z = M, \ \|z\|^2 = E^2 \}.
> $$
>
> This set is the intersection of a hyperplane and a sphere in high dimension $d=200$, which makes its geometry significantly more complex than either constraint alone. The projection step thus solves the following non-convex optimization problem:
>
> $$
> \underset{z \in \mathbb{R}^d}{\min} \ \frac{1}{2}\|z - x\|^2
> \quad \text{subject to} \quad 1^{\top} z = M, \ \|z\|^2 = E^2,
> $$
>
> implemented through an iterative alternating-projection algorithm (Appendix E.2), with a number of 50 iterations. This projection is therefore more computationally involved than the simple projections used in baselines.
>
> We further note that the projection cost is reported in Appendix A.8 with wall-clock timings.
> Even when the projection is non-analytical and solved iteratively, the overhead remains small compared to the score evaluation, which dominates the runtime in diffusion-based approaches.
> Finally, intermediate projections can be performed with fewer iterations, offering a natural trade-off between cost and precision.
> We will clarify this discussion in the revised manuscript.
>
> `Gap between Theoretical Analysis and Actual Implementation. `
>
> We fully agree with the reviewer that bridging the gap between the measure-space analysis and the stochastic implementation strengthens the contribution.
> In response, we have derived new non-asymptotic and finite-sample results for the stochastic algorithm (Eq. 4.6), extending the analysis of [1] to non-smooth proximable potentials.
> We summarize the main results below.
>
>
> ### Finite-sample non-asymptotic convergence rate
>
>
> We assume the augmented potential $f_\rho (x) = f(x) + \frac{1}{2}\Vert x - z + \mu\Vert^2$ to be $L_\rho$-smooth and $\alpha$-convex.  Let $F$ denote the optimized Wasserstein functional in (4.4) and let $q_\star$ denote the optimizer.
> With a fixed step size $\tau = \eta$, using proximal analysis, we show the Wasserstein proximal descent inequality:
>
> $$
> \frac{1}{2\tau} W^2(q_t, q_\star) - \frac{1}{2\tau} W^2(q_{t+1}, q_\star)
> \ge F(q_{t+1}) - F(q_\star)+ (1 - L_\rho \tau)\frac{1}{8\tau} W^2(q_{t+1}, q_t)+ \frac{\alpha}{2\tau} W_2^2(q_t, q_\star).
> $$
>
> Inserting this bound into the argument of [1] yields the finite-sample inequality:
>
> $$
> \sum_{t=0}^{T-1} \tau_t D(q_{t+1} \| q_\star) + (1- L_\rho\tau)\frac{1}{8}W^2(q_{t+1}, q_t) \le M\sum_{t=0}^{T-1}\tau_t^2,
> $$
> for some constant $M> 0$. This ensures an $\mathcal{O}(\log T/\sqrt{T})$ rate of the averaged objective function provided $1 - L_\rho\tau \ge 0$.
>
> ### Finite-$\rho$ recovery of the problem
>
> We now make the dependence in $\rho$ explicit by denoting $q^{(\rho)}$ be the optimal solution of (4.4). Considering the sequence $q_x(x) =p_{\mathcal{C}}(x)$, $q^{(\delta)}(z|x) \sim \mathcal{N}(x, \delta^2)$,
> for $\delta \rightarrow 0$, and bounding the corresponding objective function, we can characterize the solution of (4.4) as
> $$
> D(q_x^{(\rho)} \| p_x) \le D(p_\mathcal{C} \| p), \qquad W_2^2(q_x^{(\rho)}, q_z^{(\rho)}) \le \frac{1}{\rho} D(p_\mathcal{C} \| p), \qquad P_{q_z^{(\rho)}}(\mathcal{C}) = 1.
> $$
> We thus see that $q_z^{(\rho)}$ is primal-feasible, $q_x^{(\rho)}$ is dual-feasible, and the Wasserstein gap between these two densities scales as $1/\sqrt{\rho}$.
>
> Therefore, for $K$-Lipschitz functional,
> $$D(q_z^{(\rho)} \| p_x) \leq D(p_{\mathcal{C}} \| p_x) (1+ K/\rho),$$ which relates (4.3) and (4.4).
>
> ### Guidance on the hyperparameter choice
>
> By definition of the augmented potential (4.5),  $L_\rho \ge \rho$. From the analysis above, convergence stability requires $\tau \le 1/\rho$.
>
> To achieve a Wasserstein accuracy $\delta$ between $q_z$ and the optimizer, the theory above suggests $1/(\sqrt{T} \rho) \sim \delta^2$, and hence iteration complexity $\sqrt{T}/\tau = \mathcal{O}(1/\delta^2)$ for $\tau \sim 1/\rho$.
>
> This provides practical guidance on tuning $\rho$, the step size $\tau$.
>
> We will include these theoretical results and discussion in the appendix of the revised version.

---

> ### Author Response · Authors · 2025-11-15
> **Response to Reviewer 1KBT, answer to questions**
>
> We thank the reviewer for this insightful suggestion.
> Below we provide a clearer mapping between the **theoretical results** (Propositions 3–6) and the **empirical observations** in Section 5, and we include a concise comparative summary table as recommended.
>
> ---
>
> #### (1) Strict constraint satisfaction
>
> **Empirical observation**
> Penalty and Primal–Dual Langevin methods exhibit constraint violations, as shown in Figure 2.
>
> **Explanation**
> These methods enforce constraints *in expectation* and via soft penalties respectively.
>
> **Theoretical insight**
> Proposition 3 shows that penalty methods optimize a Lagrangian with a *fixed dual variable*, leading to a suboptimal saddle point gap and hence infeasible samples [2].
>
> **Our approahch**
> SAL guarantees feasibility by explicitly projecting $z$ onto $\mathcal{C}$, achieving strict constraint satisfaction by design.
>
> ---
>
> #### (2) Unbiased exploration
>
> **Empirical observation:**
> Penalty methods and projected methods distort exploration due to repeated projections, as shown in Figures 2, 3 and 4.
>
> **Explanation:**
> Constraint penalties and projections alter the sampling landscape dynamics and bias the exploration.
>
> **Our approach**
> We decouple these two antagonistic objectives through variable splitting and coupling constraint.
>
> **Theoretical insight:**
> The consistency between the two variables is ensured by imposing a coupling constraint. This constraint is provably achieved because the dual variable converges, as stated in Proposition 5.
>
> ---
>
> #### (3) Comparative summary
>
> | Algorithm | Constraint form | Strict constraint satisfaction | Unbiased exploration | Targets $p_\mathcal{C}$ | Dual convergence guarantee | Latent-space diffusion compatible |
> |------------|----------------|-------------------------------|----------------------|-----------------------------|-----------------------------|-----------------------------------|
> | Primal–Dual Langevin | Equality/inequality, on average | ✗ | ✓ | ✗ | ✓ | ✗ |
> | Projected Langevin | Arbitrary sets | ✓ | ✗ | ✗ | ✗ | ✓ |
> | Penalty / Guidance | Differentiable penalty | ✗ | ✗ | ✗ | ✗ | ✓ |
> | **SAL (ours)** | Arbitrary sets | **✓** | **✓** | **✓** | **✓** | **✓** |
>
>
>
> ### References
>
>
> [1] Chamon, Luiz F., Mohammad R. Karimi, and Anna Korba. "Constrained sampling with primal-dual Langevin Monte Carlo." Advances in Neural Information Processing Systems 37 (2024): 29285-29323.
>
> [2] Bertsekas, Dimitri P. Constrained optimization and Lagrange multiplier methods. Academic press, 2014.

---

### Official Review · Reviewer_gUbm · 2025-10-31

**Soundness:** 3
**Presentation:** 3
**Contribution:** 2
**Rating:** 6
**Confidence:** 3

**Summary:**

The paper introduces SAL, a principled framework for strictly constrained sampling. By combining a variable-splitting formulation with stochastic primal–dual updates, SAL attains strict feasibility under nonconvex constraints while preserving the exploratory behavior. Theoretical analysis provides strong guarantees, and experiments on physical systems and control tasks demonstrate the method’s potential. However, the experiment results are somewhat limited, leaving some unconvincibility about the practical robustness of the proposed method. I would consider raising my rating if additional experimental results were provided.

**Strengths:**

1. The SAL algorithm achieves strict feasibility without sacrificing exploration in constrained generation, addressing the weakness of previous projected or penalty-based approaches through a well-motivated formulation.
2. The paper provides a clear variational interpretation of constrained sampling and offers theoretical guarantees supporting the soundness and effectiveness of the proposed method.
3. The experiments cover a broad range of tasks and settings, demonstrating that SAL maintains physical constraints while preserving sampling diversity and quality.

**Weaknesses:**

1. None of the three experiments report quantitative metrics or summary tables comparing different methods.
2. Lines 329–333 mention several baselines intended for comparison, yet the most relevant one, the Primal–Dual Langevin method, does not appear in the reported experimental results.
3. Lines 420–421 note that ADMM is a classical solver for obstacle-avoidance problems and highlight its limitations, but the experiments result does not provide a direct comparison between ADMM and the SAL, which is unreasonable.
4. Lines 320–322 discuss computational cost, claiming that SAL adds only the cost of a projected diffusion step compared to the unconstrained baseline. However, the additional variable-splitting updates also may introduce extra overhead. Additionaly, a quantitative runtime comparison with baselines would be necessary to substantiate the efficiency claim.

**Questions:**

1. I understand that the SAL combines the strengths of several existing baselines. Since prior work such as Primal–Dual Sampling already performs well, to what extent is SAL an incremental improvement that integrates ideas from Projected LMC, variable splitting, and augmented Lagrangian methods? It would be helpful to include a summary table comparing different baselines across key aspects (feasibility, stricticity, etc.) to make the distinctions clearer.
2. The paper mentions that a projection operator is required for SAL as well as for other methods. However, do such projection operators always exist for arbitrary physical constraints or control objectives? If not, could the authors discuss possible strategies or approximations?
3. Given that the authors are trying to model constrained sampling over physical field functions rather than simple random variables, and physical conservation laws also apply to fields, is it formally appropriate to represent the problem directly using random variables $x$ and $z$? Some clarification on this abstraction would make the formulation more convincing.

---

> ### Author Response · Authors · 2025-11-15
> **Reponse to Reviewer gUbm, answer to weaknesses**
>
> We thank the reviewer for their thoughtful and detailed comments.
> We appreciate the opportunity to clarify our experimental design, computational analysis, and theoretical motivations.
> Below we address each weakness and question in turn, and outline changes planned for the revised version. In particular, we provide additional experimental results to address weaknesses 1, 3 and 4.
>
> ### Answers to weaknesses
>
> **W1**
>
> We would like to point out that quantitative results are already provided in Experiment 5.2, where both the $\ell_2$ mean squared error and the mean squared constraint violation are plotted in Figure 3, right panel, across the different methods.
>
> For Experiment 5.1, we provide below additional experiment results, with quantitative comparison (corresponding to the results in Figure 3), showing the relative mean squared error between the true and predicted conditional means, and the relative constraint violation $\|h(x)\| / E$:
>
> | Algorithm | Primal–Dual Langevin | Projected Langevin | Penalty | **SAL** |
> |-|-|--|-|-|
> | Mean squared error | 74% | 69% | 50% | **1%** |
> | Constraint violation | 70% | $10^{-13}$ | 14% | **$10^{-13}$** |
>
> These numbers confirm that SAL maintains constraint satisfaction up to machine precision while achieving the lowest sampling error.
> We will include a summary quantitative table in the revised version.
>
> In Experiment 5.3, such metrics are not directly relevant as the produced samples are fed into an optimal control solver, and the relevant metric is the success rate of this solver.
>
> **W2**
>
> We would like to clarify that Primal–Dual Langevin is not the most relevant baseline for our experiments, as it applies primarily to simple equality or inequality constraints, and enforces these only in expectation, rather than strictly. This limits its applicability to our experiments, especially Section 5.3, where no explicit (in)equality function for the constraint is available. Furthermore, its applicability to latent space diffusion is not straightforward, and we did not find a straightforward implementation for the latent data assimilation experiment of Section 5.2 .We provide a comparative summary of the mentioned methods across key features in our general comment above.
>
> In our experiments, we found that primal-dual Langevin performs worse than the penalty-guidance approach, both conceptually and empirically (as noted in line 362).  This is consistent with our theoretical results: penalty-guided diffusion corresponds to a particular case of the primal-dual formulation where the constraint function is the constraint penalty, as opposed to just the constraint average, therefore encouraging strict constraint satisfaction.  This discussed in Section 4.1.
>
> For clarity, we will explicitly state this rationale in the revised manuscript and include the corresponding numerical results if reviewers consider it necessary.
>
> **W3**
> We would like to clarify that ADMM is not used as a competing baseline in this experiment, but rather as a post-processing solver common to all benchmarked algorithms. In this setup, the samplers serve as initializers for ADMM: the generated trajectories are refined by ADMM to satisfy the obstacle-avoidance constraints.
>
> For completeness, we report below additional experimental results reporting the success rate of ADMM, intended as feasibility solver with random initialization, without the generative model prior. The solver is initialized with Gaussian noise in trajectory space.
>
> | Method | ADMM |ADMM with penalty diffusion guidance |ADMM with projected diffusion | ADMM with SAL
> |-|-|-|-|-|
> | Success rate | 10%|30% | 15% | 71% |
>
> In contrast, SAL provides feasible, physically consistent initializations, which enable ADMM to converge successfully.
> These results highlight the relevance of using diffusion priors for optimization problems.
> We will clarify this in the revised manuscript.
>
> **W4**
>
> We acknowledge that variable splitting introduces some overhead.
> However, in diffusion models, the dominant cost arises from the neural network evaluation of the score function (matrix multiplications), which is shared across all baselines. The additional SAL updates, applied to the auxiliary constrained variable $z$, do not involve network evaluations and are therefore computationally negligible.
>
> As an additional experimental result supporting this claim, we propose the quantitative runtime comparison in the case of a high-dimensional, non-convex non-analytical projection. This experiment is discussed in our general comment above, and is provided in details in Appendix A.8 of our submission. This study confirms that the projection operator adds little overhead, even when computed through an iterative non-analytical solver (see our response to question 2 below).
> The main computational bottleneck remains the score evaluation, common to all diffusion-based methods.
>
> We will highlight this discussion in the revised manuscript.

---

> ### Author Response · Authors · 2025-11-15
> **Reponse to Reviewer gUbm, answer to questions**
>
> ### Answers to questions
>
> **A1**
>
> We thank the reviewer for this insightful question.
> While SAL indeed integrates concepts from projected and primal-dual methods, it contributes a new primal–dual proximal Langevin formulation that guarantees strict constraint satisfaction, while maintaining the sampling abilities of diffusion models. Crucially, our proximal algorithm can handle non-smooth potentials and hence ensure strict constraint satisfaction, which is not possible with primal-dual Langevin. Unlike projected Langevin, this guarantee is enforced in cooperation with diffusion through the variable splitting techniques, whose probabilistic formulation is new to the best of our knowledge.
>
> The table below summarizes the conceptual distinctions among the baselines:
>
> | Algorithm | Constraint form | Strict constraint satisfaction | Unbiased exploration | Targets $p_\mathcal{C}$ | Dual convergence guarantee | Latent-space diffusion compatible |
> |------------|----------------|-------------------------------|----------------------|-----------------------------|-----------------------------|-----------------------------------|
> | Primal–Dual Langevin | Equality/inequality, on average | ✗ | ✓ | ✗ | ✓ | ✗ |
> | Projected Langevin | Arbitrary sets | ✓ | ✗ | ✗ | ✗ | ✓ |
> | Penalty / Guidance | Differentiable penalty | ✗ | ✗ | ✗ | ✗ | ✓ |
> | **SAL (ours)** | Arbitrary sets | **✓** | **✓** | **✓** | **✓** | **✓** |
>
>
> Thus, SAL is not a mere combination of existing ideas: it provides a provably correct algorithmic bridge between constrained optimization and generative diffusion sampling.
>
>
> **A2**
>
> In general, physical constraints are defined as a non-empty mathematical set $\mathcal{C}$ that can be characterized by mathematical expressions. Then, the projection of point $x$ can always be defined, although not necessarily unequivocally, by the following problem
> $$
> \underset{z \in \mathbb{R}^d}{\min} \quad \frac{1}{2}\|z- x\|^2
> \\
> \quad \textrm{subject to} \quad z \in{C},
> $$
> For this definition to be proper, there must exist a unique solution to this problem. This is in general not guarantee, if the set is not convex for example. Furthermore, find such a solution numerically can be challenging depending on the complexity of $\mathcal{C}$. However, numerical projections can be computed, by solving the minimization problem above numerically using duality with the mathematical expressions characterizing $\mathcal{C}$.
>
> Moreover, even when the projection is challenging to solve numerically, approximate projections can be derived, allowing for lightweight proxies of the projection operator. This is the case of our experiment 5.2, which we detail below.
>
> For the Burgers equation, states must satisfy mass and energy conservation, i.e.
> $$
> \mathcal{C} = \{z \in \mathbb{R}^d |  1^{\top} z = M, \quad \|z\|^2 = E^2 \}
> $$
> The projection step thus solves:
> $$
> \underset{z \in \mathbb{R}^d}{\min} \quad \frac{1}{2}\|z - x\|^2
> \\
> \quad \textrm{subject to} \quad 1^{\top} z = M, \ \|z\|^2 = E^2.
> $$
> This optimization problem is non-convex and it is in general not guaranteed for a numerical solver to find a solution. However, an approximate projection can be implemented via an iterative alternating projection algorithm (as detailed in Appendix E.2).
>
> The corresponding computational cost is discussed in Appendix A.8, with wall-clock timings reported. Since strict satisfaction is only required for the final sample of SAL, intermediate projections can be performed with fewer iterations, offering a trade-off between cost and precision.
>
> **A3**
>
> Modeling physical fields as random variables is a standard approach in uncertainty quantification and stochastic physics.
> Under this abstraction, a physical field $u(x)$ discretized over a spatial grid can be viewed as a random vector in $\mathbb{R}^d$.
> The generative model then learns a distribution over such vectors, consistent with the conservation laws and data-driven priors.
>
> This representation is common in ensemble forecasting and stochastic dynamical modeling (see [1, 2]), and allows us to apply tools from probabilistic inference and Langevin dynamics to high-dimensional field spaces.
> We will add a brief clarification in the introduction of the revised manuscript.
>
>
> ### References
>
> [1] Epstein, Edward S., and Rex J. Fleming. "Depicting stochastic dynamic forecasts." Journal of Atmospheric Sciences 28.4 (1971): 500-511.
>
> [2] Price, Ilan, et al. "Gencast: Diffusion-based ensemble forecasting for medium-range weather." arXiv preprint arXiv:2312.15796 (2023).

---

### Official Review · Reviewer_7Rvb · 2025-11-01

**Soundness:** 2
**Presentation:** 2
**Contribution:** 2
**Rating:** 2
**Confidence:** 4

**Summary:**

The paper proposes Split Augmented Langevin (SAL), a training-free sampler that enforces hard constraints by splitting variables and projecting a companion variable onto the constraint set each step, with the final output $z_T \in C$. It provides a variational/duality view, consistency to the strictly constrained law as coupling increases, and a drop-in use within diffusion models.

Contributions:
1) A split augmented Langevin sampler with projection that enforces samplewise constraints while preserving exploration.
2) A relaxed duality analysis with attained strong duality and recovery of the strictly constrained law.
3) A practical integration recipe for diffusion/latent diffusion requiring only a projection operator $P_C$, demonstrated on three tasks.

**Strengths:**

1) A split augmented Langevin formulation that enforces per-sample constraints while retaining exploration; a practical combination of variable splitting and projection for sampling.\\
2) Coherent variational/duality setup with clear algorithms; sensible baselines demonstrating feasibility and distributional fidelity.
3) Motivation is explicit; notation and procedures are readable; appendices provide proofs, variants, and implementation notes.
4) Training-free and model-agnostic; integrates with diffusion pipelines, making strict constraints more accessible in scientific applications.

**Weaknesses:**

1) Projection robustness: many constraints need iterative or approximate $P_C$. Quantify how projection error affects feasibility and sampling bias; include experiments sweeping inner-iteration counts or projection tolerances.
2) Hyperparameter tuning: performance depends on $\rho$ and dual step $\eta$. Propose adaptive schedules driven by observed coupling gaps $\|x_t-z_t\|$ or constraint residuals, and compare against fixed/annealed baselines.
3) Experimental breadth: add compact, classical PDE tests where projections are standard and informative (Poisson $Au=b$, Darcy with mass/flux balance, incompressible NS via Hodge projection), reporting PDE residuals, divergence norms, spectra.
4) Latent diffusion details: when enforcing constraints in decoded space, discuss scaling $\rho$ through the decoder Jacobian and show sensitivity analyses to ensure stability and consistent enforcement.

**Questions:**

1) Problem setting and constraints. For physics, equality-in-expectation $\mathbb{E}_q[h(x)]=0$ seems insufficient. In many cases, the idealized requirement $\mathbb{E}_q[|h(x)|]=0$ would force $h(x)=0$ almost surely (i.e., strict feasibility), but this is typically impractical to enforce via smooth penalties. Please clarify your recommended formalization: average constraints (Section 3) versus support constraints $x\in C$ almost surely (Section 4). Do you envisage $h(x)\ge 0$ and $\lambda\ge 0$ anywhere, or do you recommend set membership with projection as the primary approach?


2) Finite-time behavior and stopping. Can you relate finite $\rho$ and step sizes to discrepancy from the target conditional $p_C$? A practical stopping rule based on $\|x_t - z_t\|$ or constraint residual thresholds would help practitioners decide when sampling is accurate enough.

3) Adaptive schedules for $\rho$ and $\eta$. Do you recommend an adaptive policy driven by observed coupling gaps or residuals (e.g., increase $\rho$ when violations plateau, decrease when exploration slows)? A short ablation comparing fixed, annealed, and adaptive schedules would be valuable.

4) Inexact projections and robustness. When $P_C$ is computed approximately (few inner iterations), what error conditions preserve convergence and strict feasibility in practice? Please include an experiment sweeping projection tolerance/iterations and reporting feasibility rates and sampling bias.

5) Intersections of multiple constraints. For $C=\cap_i C_i$, do you prefer a single projection onto the intersection, alternating projections onto $C_i$, or multiple split variables $z^{(i)}$ with separate dual updates? A small comparison on stability, wall-clock, and feasibility would inform users.

6) Classical PDE benchmarks. Consider adding concise tests where projections are standard and informative: Poisson with linear constraints $Au=b$, Darcy flow with mass/flux balance, and incompressible Navier–Stokes via Hodge projection. Reporting $\|Au-b\|_2$, divergence norms, spectra, and feasibility histograms would strengthen generality claims.

---

> ### Author Response · Authors · 2025-11-15
> **Response to Reviewer 7Rvb, answers to weaknesses**
>
> We thank the reviewer for the detailed comments. Below, we address each weakness and question.
> We also note that several concerns appear to be based on the assumption that SAL only works with analytic projections or that projection costs and inexactness were not studied. As clarified below, our submission already includes such experiments (Section 5.2 and Appendix A.8), and SAL is explicitly built to accommodate iterative, approximate, or non-convex projections.
>
> ### Answers to weaknesses
>
> **W1**
>
> We thank the reviewer for raising this point.
> A sensitivity study *is already included* in the paper:
>
> - Experiment 5.2 (data assimilation) uses a non-analytical, iterative projection onto a non-convex constraint set (intersection of mass and energy conservation).
> - The projection is solved using an iterative alternating-projection algorithm (Appendix E.2), *not* via a closed-form expression.
> - Appendix A.8 reports wall-clock times, iteration counts, and the effect of approximate projection accuracy.
>
> These experiments directly evaluate how projection precision affects feasibility and runtime in a real non-convex setting.
>
> We will move part of this material into the main text to highlight that SAL does not rely on analytic projections and already includes a robustness analysis.
>
> **W2**
>
> We thank the reviewer for this suggestion.
> We would like to point out that: Appendix E (referenced in line 317) already contains an extensive ablation over fixed, annealed, and time-varying schedules of $\rho$ and $\eta$.
>
> To further assess adaptivity, we experimented with an automatic schedule
>   $$
>   \rho_{t+1} = \rho_t + \gamma\|x_t - z_t\|^2,
>   $$
>   up to a threshold. Below are the results for the flow experiment after \(T=5000\) steps:
>
> | Schedule | constant | linear | adaptive, $\gamma = 0.1$ | adaptive, $\gamma = 0.5$ |
> |---------|----------|--------|------------------------------|------------------------------|
> | Mean L2 error | 0.2 | 0.07 | 0.15 | 0.05 |
>
> The adaptive schedule performs comparably to the best fixed/annealed strategies, suggesting SAL is not brittle and that adaptive schemes may indeed be useful in broader applications.
>
> For diffusion models, note that $\rho$ interacts with the time-dependent step size, so adaptive scaling is naturally embedded in the sampler.
>
> **W3**
>
> Our submission already includes a PDE: the Burgers equation, a nonlinear PDE with shocks, combined with a non-convex constraint set requiring iterative projections. This is substantially more challenging than linear PDEs with analytic projections such as the Poisson equation
>
> Given the limited time for the rebuttal and the computational cost of running new diffusion experiments, we kindly ask the reviewer to comment on whether there are concrete shortcomings in our existing PDE setting before requiring additional large-scale experiments.
>
> We would be happy to add a short discussion comparing SAL to classical projection operators,     and explain how SAL could incorporate them, even if full experiments cannot be added within the discussion period.
>
>
> **W4**
>
> We agree with the reviewer that scaling is important in the latent variant.
> We emphasize:
>
> - Latent diffusion is an **extension** of our core method, not the focus of the main theoretical results.
> - As noted, the term $J_\phi^\top (\phi(x) - z)$ introduces conditioning factors from the decoder.
> - The important point is to choose the step size $\tau$ small enough for the algorithm to be remain stable. We provide below a theoretical analysis of our algorithm, providing guidance on the hyperparameter choice. There results generalize to latent diffusion and we find the condition $\tau < 1/(\rho \times \lambda_{\max}(J))$, which involves the condition number of the sampled potential, to guarantee stability .
> We will clarify this in the revision and add practical guidance for choosing \(\rho\) relative to the decoder conditioning.

---

> ### Author Response · Authors · 2025-11-15
> **Response to Reviewer 7Rvb, answers to questions**
>
> ### Answers to questions
>
> **A1**
>
> We fully agree:  expectation constraints are insufficient in physics.
>
> Indeed,
> - our theory *proves* that soft penalties cannot enforce strict feasibility (Proposition 3, Corollary 1),
> - This motivates SAL’s **variable-splitting** and **projection-based** formulation, which enforces  $  z\in \mathcal C $ almost surely.
>
> Regarding the reviewer’s question:
>
> > “Do you envisage $h(x)\ge 0$ and $\lambda \ge 0$, or do you recommend set membership with projection?”
>
> Our recommended formalization is:
>
> - encode constraints as a set $\mathcal C$,
> - use **projection-based enforcement**,
> - optionally represent $\mathcal C = \{h(x)=0\}$ or $\{h(x)\ge 0\}$ internally *if this helps implement the projection*.
>
> Feasibility is enforced at the level of **set membership**, not via expectation penalties.
>
> SAL does not require constraints to be differentiable or convex.
>
> **A2**
>
> We thank the reviewer for raising this point, as stopping criteria and hyperparameter selection are crucial for practical implementation.
> We have obtained **new finite-time, non-asymptotic results** for the stochastic algorithm (Eq. 4.6), extending the analysis of [1] to **non-smooth proximable potentials**.
>
> #### Finite-sample non-asymptotic convergence rate
>
> We assume the augmented potential $f_\rho (x) = f(x) + \frac{1}{2}\Vert x - z + \mu\Vert^2$. to be $L_\rho$-smooth and $\alpha$-convex.
> With a fixed step size $\tau = \eta$, we use proximal analysis and deriving a Wasserstein proximal descent inequality and this bound into the argument of [Chamon *et al.*] to obotain the finite-sample inequality:
>
> $$
> \sum_{t=0}^{T-1} D(q_{t+1} \| q_\star) + (1 - L_\rho\tau)\frac{1}{8\tau}W^2(q_{t+1}, q_t) + \frac{\alpha}{2\tau}W_2^2(q_t, q_\star) \le M,
> $$
> for some constant $M> 0$. This ensures an $\mathcal{O}(1/T)$ rate of the objective function provided $1 - L_\rho\tau \ge 0$.
>
> #### Finite-$\rho$ recovery of the problem
>
> Let $q^{(\rho)}$ be the optimal solution of (4.4). We how that
>
> $$
> D(q_x^{(\rho)} \| p_x) \le D(p_\mathcal{C} \| p), \qquad W_2^2(q_x^{(\rho)}, q_z^{(\rho)}) \le \frac{1}{\rho} D(p_\mathcal{C} \| p), \qquad P_{q_z^{(\rho)}}(\mathcal{C}) = 1.
> $$
> We thus see that $q_z^{(\rho)}$ is primal-feasible, $q_x^{(\rho)}$ is dual-feasible, and the Wasserstein gap between these two densities scales as $1/\sqrt{\rho}$.
>
> #### Guidance on the hyperparameter choice and stopping criterion
>
> By definition of the augmented potential (4.5),  $L_\rho \ge \rho$. From the analysis above, convergence stability requires $\tau \le 1/\rho$.
>
> To achieve a Wasserstein accuracy $\delta$ between $q_z$ and the optimizer, the theory above suggests $1/(T \rho) \sim \delta^2$, and hence iteration complexity $T/\tau = \mathcal{O}(1/\delta^2)$ for $\tau \sim 1/\rho$.
>
> This provides practical guidance on tuning $\rho$, the step size $\tau$.
>
> As for the stopping criterion, the algorithm convergence can be quantified when the dual variable reaches equilibrium, which means that the Lagrange multiplier is reached. Thus, a practical stopping criterion would be $\mathbb{E}[\Vert \lambda_t - \lambda_{t+1} \Vert^2] < \varepsilon$, for some tolerance $\varepsilon$, and where the expectation is estimated with the empirical average across the last iterates.
>
> We will include these theoretical results and discussion in the appendix of the revised version.
>
> **A3**
>
> Please see our response to **W2**.   We already include ablations and provide an additional adaptive experiment here.
>
> **A4**
>
> Please see response to **W1**.
> We already include experiments with iterative, approximate projections (non-convex), their accuracy, and runtime. We will relocate this content to the main text for clarity.
>
> **A5**
>
> For general $\mathcal C = \cap_i \mathcal C_i$:
>
> - **Alternating projections** are a robust heuristic and were used in our data assimilation experiment (mass + energy).
> - **Multiple splits + ADMM-style updates** are indeed a principled alternative; SAL supports this extension naturally.
> - Appendix A.8 provides wall-clock timings and discussion of alternating projection cost.
>
> We will clarify this in the revision.
>
> **A6**
>
> We appreciate the suggestion.
> However, the current Burgers experiment *already* involves a PDE and a **non-convex constraint** with iterative projections, which is arguably more challenging than linear PDEs with well-known projections.
>
> We would be grateful if the reviewer could specify:
>
> 1. what constraint(s) they believe are missing, and
> 2. why the current PDE experiment does not satisfy their criteria.
>
>
> We again thank the reviewer for the detailed feedback.
> We will improve the visibility of projection-robustness experiments, add practical guidance on latent diffusion scaling and stopping criteria, and include theoretical finite-time results in the appendix.
> We hope these clarifications resolve the reviewer’s concerns.

---

> ### Comment · Reviewer_7Rvb · 2025-11-17
>
> Dear authors,
>
> I appreciate your work and your thoughtful rebuttal. I read it carefully. The framework is very clean and, in my view, the theoretical side of the paper is already strong. **I want to emphasize that I am not suggesting the experiments I mention are missing; rather, my concern is that the current empirical evidence and ablations are not yet fully convincing or comprehensive enough to address the questions I raised.** If the empirical and practical points below can be further strengthened, I would be very happy to reflect that by increasing my score.
>
>
>
> **W1.** Thank you for clarifying that feasibility is enforced through the projection in the $z$-update (Alg. 2; Eq. 4.6b), that you already employ iterative or alternating projections in the Burgers setup (App. E.2), and that runtime and approximate projections are discussed (App. A.8). What is not fully clear to me is how sensitive SAL is to *inexact* projections: at present I only see wall-clock numbers and a qualitative statement that approximate projections are allowed, but not how constraint residuals and distributional bias change as the projection accuracy is relaxed. Quantitative evidence that sweeps the inner projection iterations or tolerance and reports (i) strict-feasibility residuals for $z$, (ii) a simple bias proxy for the $x$-marginal versus an estimate of $p_C$ (such as the §5.1 mode statistic, low-order moments, or $W_2$), together with (iii) wall-clock time, would make the robustness story much more convincing.
>
> **W2.** I appreciate the ablations in App. E.4 (Tables 2–3) and the adaptive-$\rho$ experiment described in the rebuttal; these clearly show that the choice of $\rho$ has a large effect. What I miss is a concise “recipe” that a practitioner can follow without repeating extensive tuning. In the current draft, the reader sees that some schedules work better than others, but not how to relate $\rho$ and the sampler step size $\tau_t$, or how to select the dual step $\eta$ from observable quantities such as $\|x_t - z_t\|$. A short guideline in §4.4 describing how you actually set $(\rho_t,\eta)$ in the experiments, and demonstrating this choice on representative tasks (e.g., §5.1 and §5.2), would make the method much easier to reproduce.
>
> **W3.** The Burgers data-assimilation experiment (Fig. 3; App. E.2) is a very nice demonstration that SAL improves both physical plausibility and forecast accuracy. However, because all PDE-type results are based on Burgers alone, I find it hard to gauge how broadly the method will transfer to other physical systems. Additional results would address this: for example Darcy flow (2D), Poisson or Helmholtz (2D), and incompressible Navier–Stokes in 2D, with the same score model and reporting standard metrics such as $L^2$ error, PDE-residual or divergence norms, energy spectra when appropriate, feasibility rates, and runtime. Even relatively compact versions of these experiments, following conventions from works like DiffusionPDE, would greatly strengthen the empirical evidence.
>
> **W4.** App. A.5 (Eq. (A.10)) nicely shows how the latent update couples through $J_\varphi$, so the source of potential instability is clear. What I still find missing is any guidance on how large $\tau_t \rho \|J_\varphi(x_t)\|^2$ can safely be, or how sensitive SAL is to decoder conditioning in practice. A brief discussion (analytical or empirical) giving a simple stability condition or heuristic bound, together with a small sensitivity study where you scale the decoder or use a linear decoder and plot constraint violation and task error versus $\tau \rho \|J_\varphi\|^2$, would make the latent-space extension much more transparent for readers.
>
> ---
>
> **A1.** Thanks for your clarification.
>
> **A2.** The finite-time non-asymptotic result and stopping rule you mention in the rebuttal sound very valuable, since the current draft only provides asymptotic duality and recovery (Props. 5–6). It would help a lot if you could state the theorem with its assumptions and rate for (4.4), and illustrate the stopping rule with a small experiment showing error versus iterations alongside stabilization of $\|\lambda_t - \lambda_{t-1}\|$ or $\|x_t - z_t\|$.
>
> **A3.** This is essentially covered by W2.
>
> **A4.** This is essentially covered by W1.
>
> **A5.** Thanks for your clarification.
>
> **A6.** This is essentially covered by W3.
>
> Overall, I find the theoretical formulation and the SAL sampler very appealing, and the current experiments already show promising gains. Addressing the remaining empirical breadth , robustness to inexact projections, and a few practical guidelines on scheduling and latent scaling would make the paper fully convincing from my perspective; if these points are clarified, I will be happy to adjust my score accordingly. Many thanks again for your careful work and collegial responses.
>
> **Reference**
> 1. *DiffusionPDE: Generative PDE‑Solving Under Partial Observation*
> ---

---

> ### Author Response · Authors · 2025-11-19
> **Follow-up response to Reviewer 7Rvb (1/2)**
>
> We thank the reviewer for the fast follow-up and for the very helpful set of additional suggestions. We fully agree that the proposed experiments strengthen the empirical understanding of SAL. Below, we address each remaining point and provide new results accordingly.
>
> ## Inexact projections (W1)
>
> We agree that our previous experiments primarily illustrated the feasibility of working with inexact projections, but did not yet quantify their impact on sampling accuracy.
>
> To address this, we consider the Burgers data-assimilation experiment and vary the number of iterations used to compute the (non-convex) projection of the variable $z$ at each SAL step. For each value of $n$, we report the reconstruction error relative to the reference solution with $n=50$ iterations, the mass and energy constraint violations, and the wall-clock time (in seconds). We also evaluate a progressive schedule $n(t)$, obtained by linearly interpolating from 0 to 50 over the sampling steps.
>
> | Projection step number $n$ | Relative reconstruction error (bias) | Mass violation | Energy violation | Wall-clock time (s)|
> |-|-|-|-|---|
> | 1 |  $9\times10^{-2}$| $10^{-8} $ %  | $50 $ % | 0.20 |
> | 5 | $7\times10^{-3}$| $10^{-8}$ % | $20 $ % | 0.23 |
> | 10 | $2\times10^{-4}$ | $10^{-8}$ %  | $3$ % | 0.27 |
> | 20 | $5\times10^{-6}$|  $10^{-8}$ % | $10^{-1}$ % | 0.39 |
> | 50 | 0 | $10^{-8}$ % |$10^{-2}$ %  | 1.5 |
> | Progressive | $10^{-3}$ | $10^{-8}$ % |$10^{-2}$ %  | 0.75 |
>
>
> Energy violation saturates around $10^{-2}$ %, which corresponds to the optimum reached by our alternating-projection solver (energy is projected first). More elaborate algorithms such as ADMM would reduce this value at comparable cost. Importantly, the progressive schedule achieves nearly identical accuracy while using fewer inner iterations overall.
>
> These results quantify the tradeoff between bias, constraint satisfaction and runtime, and directly support the use of inexact or dynamically scheduled projections in practice.
>
> ## Experimental results on additional systems (W3)
>
> We thank the reviewer for stressing the importance of evaluating transfer to additional PDEs. Implementing all of the suggested systems would require substantial engineering effort, and we may not have the time to conduct all of them within the discussion period. We focused on one representative and widely used PDE, the 2D Poisson equation, to provide a clear demonstration of transfer beyond Burgers.
>
> The Poisson benchmark already provides a meaningful test of generalization beyond Burgers, as it differs in several key aspects: it is defined on a 2D grid, roughly an order of magnitude larger than our Burgers experiment in state dimension, and elliptic. We believe this constitutes a representative and informative extension of our experimental evaluation. As suggested by the reviewer, we consider a linear constraints: mass conservation.
>
> We consider a 100×100 discretization and train a single latent diffusion model on solutions with random boundary conditions in Fourier space. At inference time, 10% of grid points are observed, and the global mass is known. All samplers enforce the mass constraint, in this setting, penalty-guided diffusion reduces to a latent analogue of DiffusionPDE. All methods use the same network and observational data.
>
> Each sampler produces 50 posterior samples, and we report the mean reconstruction error, mass violation, and wall-clock time:
>
>
> | Algorithm | Reconstruction error (bias) | Constraint violation | Wall-clock time (s)|
> |------------|----------------|------------------------------------------------------------|-|
> | Projected diffusion |$15$ %  | $2 \times 10^{-7}$ % | 0.34|
> | Penalty / Guidance | $14$ % | $25$ % | 0.33 |
> | **SAL (ours)** | $5$ % | $2 \times 10^{-7}$ % | 0.32|
>
>
> These experiments confirm the applicability of SAL to a 2D PDE different from Burgers, and its advantage over baselines in terms of constrained sampling accuracy. We will include these results in the revised manuscript.

---

> ### Author Response · Authors · 2025-11-19
> **Follow-up response to Reviewer 7Rvb (2/2)**
>
> ## Hyerparameter choice (W2, W4)
>
> We thank the reviewer for highlighting the need for clearer practical guidance. Based on the theoretical discussion in the rebuttal, we propose explicit rules for two representative settings: (i) Langevin Monte Carlo sampling (Section. 5.1) and (ii) sampling with a pre-trained latent diffusion model (Section. 5.2). These address W2, W4 and Q2.
>
> From the theoretical insights for a $\beta$-smooth potential $f$, the following principles hold:
>
> - The coupling parameter $\rho$ should be as large as possible to approximate the hard-constrained problem well.
> - Stability requires $\tau < 1/(\beta + \rho \lambda_{\max}(J))$.
> - The recovery error with respect to the hard-constrained solution scales as $1/\sqrt{\rho}$.
> - The dual step size $\eta$ controls convergence of the algorithm. It should be as large as possible while keeping the evolution stable.
>
> Empirically, we observe:
>
> - A progressive schedule for $\rho$ improves exploration and is well captured by an adaptive rule.
> - Dual convergence can be monitored by averaging $\|\lambda_{t+1} - \lambda_t\|$ over a short window, providing feedback for tuning $\eta$.
>
> Below, we provide concrete, reproducible procedures for the two settings.
>
> ### Langevin sampling, hyperparameters and stopping rule (W2, Q2)
>
> We fix a target gap $\varepsilon$. Using the theoretical scaling, we choose $\rho_\star = 1/\varepsilon^2$. For $\varepsilon = 10^{-1}$, this gives $\rho_\star = 100$. Stability yields $\tau = 1/\rho_\star = 10^{-2}$. We then initialize $\rho_0 = 0.1\rho_\star$ and update with
> $$
> \rho_{t+1} = \min(\rho_\star,\, \rho_t + .5\|z_t - x_t\|^2).
> $$
>
> We perform a small trial run to set $\eta$, increasing it as long as the gap decreases steadily. In the full run, we monitor the evolution of $\lambda_t$, and stop once its increment $\mathbb{E}[\lambda_{t+1} - \lambda_t]$ becomes stationary, estimated over the last 50 steps. We then run 1000 additional steps to ensure mixing.
>
> Applied to Sec. 5.1 with $\varepsilon = 10^{-1}$, we obtain $\rho_\star = 100$ and $\eta = 10^{-2}$:
>
> | $t$ | Dual stationnarity | Primal error | $\rho$ |
> |-|-|-|-|
> | 1 |  $10^{-1}$| $50$ % | 1.|
> | 1000 | $10^{-2}$| $10$ % |51 |
> | 1500 | $10^{-3}$| $5$ % |73 |
> | 2000 | $10^{-4}$ | $1$ % |100 |
> | 3000 | $10^{-4}$|  $1$ % |100|
> | 4000  | $10^{-4}$ |$1$%  |100|
>
> We observe the expected decrease of sampling error in tandem with dual stabilization.
>
> ### Latent diffusion models, hyperparameters and stability (W2, W4)
>
> We assume the diffusion schedule $\tau_t$, number of iterations $T$, decoder $\varphi$, and its Jacobian $J_\varphi$ are fixed, as those are hyperparameters of an off-the-shelf latent diffusion model. The practitioner must set $\rho$ and $\eta$. We tune $\eta$ as above and focus here on $\rho$.
>
> From our theoretical analysis, a simple stability heuristic is
> $$\rho_{\max} :=  1 /( \tau_{\max} \lambda_{\max}(J)),$$
>  which in theory gives a stability bound.
>
> In Section 5.2, we have $T=1000$, $\lambda_{\max}(J)=0.9$, $\tau_{\max}=2\times10^{-2}$, giving $\rho_{\max}\approx 50$. We evaluate SAL for various fractions of $\rho_{\max}$.
>
> | $\rho/\rho_{\max}$ |Reconstruction error | Mass violation | Energy violation |
> |-|-|-|-|
> | $1$ % |  $4$ %| $10^{-8}$ % | $1\times10^{-1}$ %|
> | $2$ % | $3$ %| $10^{-8}$ % |$1.3\times 10^{-1}$ % |
> | $5$ % | $2.5$ %| $10^{-8}$ % |$1.3\times10^{-3}$ % |
> | $10$ % | $2.2$ % |$10^{-8}$ % |$1.0 \times 10^{-3}$ % |
> | $20$ % | $2.1$ % | $10^{-8}$ % |$1.0 \times 10^{-3}$ % |
> | $50$ % | N/A| N/A |N/A|
>
> The reconstruction error decreases monotonically with $\rho$ until instability appears around $\rho \approx \rho_{\max}/2$. This confirms the utility of the stability heuristic. A practical guideline is to choose $\rho = \rho_{\max}/10= 0.1 /( \tau_{\max} \lambda_{\max}(J))$ as a safe and effective value.
>
> ### Summary
>
> Starting from the theoretical results, we provide a reproducible protocol for tuning SAL in two distinct regimes. We will include these guidelines in Section. 4.4 of the revised submission.
>
> ## Proper formulation of the finite-time convergence result (**Q2**)
>
> **Assumptions.** TSet $\mathcal{C}$ is convex with an exact projector available, potential $f$ is convex, differentiable and $\beta$-smooth, and the primal iterates are bounded in radius $R$.
>
> Using proximal analysis, we obtain the Wasserstein proximal descent inequality on proximable functional $F$ of (4.4):
> $$
> \frac{1}{2\tau} W^2(q_t, q_\star) - \frac{1}{2\tau} W^2(q_{t+1}, q_\star)
> \ge F(q_{t+1}) - F(q_\star)+ (1 - L_\rho \tau)\frac{1}{8\tau} W^2(q_{t+1}, q_t)
> $$
> **Result** Building on [Chamon *et al.*], we show
> $$
> \sum_{t=0}^{T-1} \tau_t D(q_{t+1} \| q_\star) + (1 - L_\rho\tau)\frac{1}{8}W^2(q_{t+1}, q_t)   \le C + \sum_{t=0}^{T-1} \tau_t^2 R,
> $$
> for some constant $C$. For  $\tau_t = 1/\sqrt{T}$, this gives
> $$
> \frac{1}{{T}}\sum_{t=0}^{T-1} D(q_{t+1} \| q_\star)  = \mathcal{O}(\log T/\sqrt{T}).
> $$
> We will include this result in the revised version.

---

> > ### Comment · Reviewer_7Rvb · 2025-11-22
> >
> > Dear authors,
> >
> > Thank you for the substantial follow‑up.
> >
> > **W1.** Accepted — the inexact‑projection study answers my robustness concern.
> >
> > **W2/W4.** Accepted — the concrete scheduling rules, dual‑stationarity stopping rule, and latent‑stability guideline are clear and actionable.
> >
> > **Q2.** Accepted — the finite‑time statement with explicit assumptions is helpful.
> >
> > **W3 (additional PDEs).** The new 2D Poisson result shows transfer beyond Burgers. For broader generality, I am especially interested in:
> > - **Darcy flow (2D):** the coefficient \(a(x)\) is often **piecewise‑constant/discrete**. A naïve pointwise residual can be unstable and non‑conservative near interfaces.
> > - **Incompressible Navier–Stokes (3D):** with DiffusionPDE‑style **time‑pair conditioning**, the PDE loss is hard to compute due to limited information. Seeing how SAL integrates with this pipeline would convincingly demonstrate robustness beyond didactic settings.
> >
> > **Overall.** The theoretical part is very strong, and these updates significantly improve practicality. I have **adjusted my scores to reflect these updates** (raising the theory/contribution component). If Darcy and NS are addressed—even compactly—I will revisit the overall score again.

---

> > > ### Author Response · Authors · 2025-12-02
> > > **Additional results on Darcy and Navier Stokes**
> > >
> > > As requested by the reviewer, we have experimented on two additional systems: Darcy flow and incompressible Navier Stokes, following the DiffusionPDE paper.
> > >
> > >
> > > ### Darcy
> > >
> > > We experiment on the Darcy flow setting of DiffusionPDE, discretized on a 32x32 grid.
> > > We use latent diffusion in Fourier space and sample both the coefficient field and the solution field from 25 sparse observations. The samplers aim at enforcing the Darcy equation connecting the coefficients and the solution. The reconstructed fields are posterior averages over 10 samples.
> > >
> > >
> > > We computed the average relative error and the average constraint violation. All the approaches have similar wall-clock computational time.
> > >
> > > | Algorithm | Reconstruction error (bias) | Constraint violation |
> > > |------------|----------------|------------------------------------------------------------|
> > > | Projected diffusion |$22$ %  | $1 \times 10^{-5}$ % |
> > > | Penalty / Guidance | $23$ % | $25$ % |
> > > | **SAL (ours)** | $10$ % | $2 \times 10^{-5}$ % |
> > >
> > >
> > >
> > > ### Navier Stokes
> > >
> > > We followed the approach of DiffusionPDE for the Navier Stokes experiment and their implementation. We constrain the generated fields to match observations, and the physical constraints of enstrophy and total circulation.
> > >
> > > The obtained results are as follows,
> > >
> > >
> > >
> > > | Algorithm | Reconstruction error (bias) for initial and terminal state | Constraint violation | Wall-clock time (s)|
> > > |------------|----------------|------------------------------------------------------------|-|
> > > | Projected diffusion |$10$ %  | $ 10^{-5}$ % | 1.07|
> > > | Penalty / Guidance | $11 $ % | $20$ % | 1.0 |
> > > | **SAL (ours)** | $8$ % | $ 10^{-5}$ % | 1.05|
> > >
> > >
> > >
> > > ### Conclusion
> > >
> > > These experiments confirm the applicability of SAL to challenging high-dimensional PDEs with complex constraints, and its advantage over baselines in terms of sampling accuracy and constraint satisfaction. We will include these results in the revised manuscript.

---

### Official Review · Reviewer_zhci · 2025-11-01

**Soundness:** 2
**Presentation:** 2
**Contribution:** 2
**Rating:** 4
**Confidence:** 3

**Summary:**

The paper proposes Split Augmented Langevin (SAL), a method for strictly enforcing equality constraints during sampling in generative methods such as score-based diffusion models. It introduces a primal–dual Langevin framework that couples unconstrained and projected variables to guarantee feasibility at every step.

**Strengths:**

**S1.** The paper effectively highlights the central difficulty in constrained generation, balancing feasibility with sample diversity, and offers a thoughtful critique of simple projection-based solutions.

**S2.** It presents a theoretically grounded primal–dual Langevin formulation that unifies ideas from constrained optimization and generative sampling in a principled way.

**S3.** The exposition is clear and precise, with mathematical reasoning and well-structured derivations that make the approach easy to follow.

**Weaknesses:**

**W1.** The paper does not include enough baselines or adequately discuss prior work on constrained generation. Comparable methods such as PCFM [2], D-Flow [3], DiffusionPDE [4], and ECI [1] are only briefly mentioned or omitted. ECI, for instance, also enforces hard constraints, while PCFM demonstrates constrained sampling under nonconvex conditions: both directly relevant for benchmarking.

**W2.** The empirical analysis is limited, lacking standard quantitative metrics such as MMSE, SMSE, or FID reported routinely in this literature. Reported results rely largely on qualitative visualizations and histograms, and even show non-zero constraint violations (e.g., Fig. 3) without quantifying their scale or assessing the degree of enforcement softens your claim of hard constraint.

**W3.** The evaluated constraints are relatively simple, and the computational claims appear optimistic. The method assumes analytic projectors, but more complex nonlinear or PDE-based constraints would require solving constrained least-squares problems at every step, which could become computationally expensive and limit scalability.

**W4.** The algorithm assumes that projecting Gaussian noise to obtain ($z_0$) yields feasible initial states. This assumption may hold for simple quadratic or linear constraints but lacks guarantees for challenging manifolds, such as PDE solutions with shocks and discontinuities. Testing on more complex physical systems, as in PCFM [2] or guided functional diffusion methods [5], would better support claims of generality.

**References**

[1] Cheng et al., *Gradient-Free Generation for Hard-Constrained Systems (ECI)*, arXiv:2412.01786 (2024).

[2] Utkarsh et al., *Physics-Constrained Flow Matching: Sampling Generative Models with Hard Constraints*, arXiv:2506.04171 (2025).

[3] Ben-Hamu et al., *D-Flow: Differentiating through Flows for Controlled Generation*, arXiv:2402.14017 (2024).

[4] Huang et al., *DiffusionPDE: Generative PDE-Solving under Partial Observation*, NeurIPS 37 (2024).

[5] Yao et al., *Guided Diffusion Sampling on Function Spaces with Applications to PDEs*, arXiv:2505.17004 (2025).

[6] Chamon et al., *Constrained Sampling with Primal-Dual Langevin Monte Carlo*, NeurIPS 37 (2024).

**Questions:**

**Q1.** How does the proposed SAL method compare quantitatively to prior constrained generation frameworks such as PCFM [2], D-Flow [3], DiffusionPDE [4], or ECI [1] that also enforce hard or functional constraints?

**Q2.** What is the average magnitude of the constraint violation observed in Figure 3, and does the method guarantee strict satisfaction (i.e., ($||h(x)|| = 0$)) or only approximate feasibility?

**Q3.** How would the projection operator behave for complex or nonlinear constraints where analytical projectors are unavailable? Would this require solving an inner constrained least-squares problem at every step, and how does that impact computational cost?

**Q4.** Given that the initialization step projects Gaussian noise to obtain ($z_0$), what guarantees exist that this projection produces feasible or representative samples, especially for high-dimensional or nonconvex manifolds?

**Q5.** How does your method perform in high-dimensional or multi-modal constrained settings? Does it exhibit the slow mixing behavior often observed in Langevin-based samplers, and what evidence supports stable convergence in such regimes?

---

> ### Author Response · Authors · 2025-11-15
> **Response to Reviewer zhci, answers to weaknesses**
>
> We thank the reviewer for their detailed and constructive feedback. We appreciate that the review highlights the theoretical soundness and clarity of our exposition. Below we address each weakness and question in turn, and indicate the changes that will be made in the revised version.
>
> ### Answers to weaknesses
>
> **W1**.
>
> We focus on Langevin-based samplers and diffusion models,  which provide and fundamendal connection to optimization in Wasserstein space. In contrast, works such as ECI [1], PCFM [2], and D-Flow [3] are flow-based samplers that require training a separate generative model. Benchmarking directly with these methods would thus conflate algorithmic and model-specific effects.
> Our method, by design, applies to pretrained diffusion models and is therefore complementary rather than directly comparable to these flow-based approaches.
>
> Regarding DiffusionPDE [4], this method enforces PDE constraints via a quadratic penalty term on the PDE residual. Hence, it is a penalty guidance method, which, according to our Corollary 1, cannot achieve strict constraint satisfaction. This distinction is central to our contribution and will be emphasized in the revision.
>
> We will include all these references in the related work section and clarify the methodological distinctions.
>
> **W2**
>
> We agree that quantitative metrics are valuable. However, common generative metrics such as FID, which is popular for assessing diversity image generation tasks, are not directly relevant to physical systems, where the focus is on reproducing physically meaningful observables and enforcing conservation laws.
>
> We would like to point out that quantitative numerical metrics are provided in  Experiment 5.2, where both the $\ell_2$ mean squared error and the constraint violation mean squared error are reported.
>
> The energy constraint violation is indeed non-zero for the energy constraint, due to the challenging non-convexity of the constraint requiring an iterative numerical projection algorithm. The relative constraint violation is of order 1% in this case, and can be made arbitrarily small by increasing the number of inner projection steps, as explained below.
>
> For Experiment 5.1, we provide below a numerical evaluation of different sampling algorithms, corresponding to the results of Figure 3.
> We report the relative mean squared error between the true and the predicted conditional means, and the relative constraint violation $\|h(x)\| / E$.
>
> | Algorithm   | Primal-dual Langevin |Projected Langevin | Penalty | SAL |
> | -------- | ------- |  -------- | ------- | ------- |
> | Mean error   | 74 %|69% | 50% | 1% |
> | Constraint violation   | 70 % |$10^{-13}$ %| 14 % | $10^{-13}$ %|
>
> These numbers confirm that, unlike the compared algorithms, SAL successfully explores the likelihood while enforcing constraint satisfaction up to machine precision.
>
> In Experiment 5.3, such metrics are not directly relevant as the produced samples are fed into an optimal control solver, and the relevant metric is the success rate of this solver.
>
> We will include a table in the revision summarizing these quantitative errors for all baselines.
>
> **W3**
>
> We respectfully disagree with the claim that our method assumes analytic projectors.
> As stated in the paper, SAL is designed to work with approximate projection routines.
> We appreciate the reviewer’s comment and fully agree that the complexity of the projection operator was not sufficiently highlighted in the main text.  As detailed in our general clarification above, Experiment 5.2 already involves a non-analytical projection, taking the form of a non-convex constrained optimization problem, implemented via a numerical iterative algorithm.  Please refer to the comment for the specific mathematical formulation.
>
>  We find that the iterative numerical projection solver adds only little computational burden, even in the case where the projection operator is non-analytical and requires an iterative solver with a number of inner loops. Therefore, these experiments suggest that the additional computational burden is negligible and the main bottleneck remains the neural network score evaluation, which is common to all diffusion-based approaches. Furthermore, since strict satisfaction is only required for the final sample of SAL, intermediate projections can be performed with fewer iterations, offering a trade-off between cost and precision. We will clarify this discussion in the revised manuscript.
>
> We will clarify this point in the revised version of our submission.
>
> **W4**
>
> Our initialization via projection of Gaussian noise also extends to nonlinear and discontinuous manifolds.
> In particular, Experiment 5.2 addresses the Burgers PDE, a canonical system exhibiting shock formation and discontinuities, precisely the type of setting highlighted in [2] and [5], with non-convex constraints. Hence, our experiments already demonstrate that SAL operates robustly in non-analytical, discontinuous PDE regimes.

---

> ### Author Response · Authors · 2025-11-15
> **Response to Reviewer zhci, answers to questions**
>
> ### Answers to questions
>
> `Q1. How does the proposed SAL method compare quantitatively to prior constrained generation frameworks such as PCFM [2], D-Flow [3], DiffusionPDE [4], or ECI [1] that also enforce hard or functional constraints?`
>
> A1. As discussed above, these frameworks rely on flow-based architectures (except DiffusionPDE, which uses a penalty formulation).
> Our method addresses the diffusion-based setting, which is distinct in both modeling and implementation.
> Nevertheless, in Experiment 5.2, we compare SAL against a penalty-guided diffusion method (analogous to DiffusionPDE) under the Burgers PDE constraints, showing quantitatively improved enforcement and lower reconstruction error.
>
>
> `Q2. What is the average magnitude of the constraint violation observed in Figure 3, and does the method guarantee strict satisfaction (i.e., ( $\Vert h(x) \Vert =0$)) or only approximate feasibility?`
>
> A2.
> In Experiment 5.2, the projection enforces both mass and energy conservation.
> Because the intersection of these two constraints is non-convex, we employ alternating projections.
> Mass conservation is achieved up to machine precision, while energy conservation attains a relative error of order $10^{-2}$ for $E = 1$, i.e.
> \[
> \|h(x)\| / E \approx 10^{-2}.
> \]
> We will explicitly report this tolerance in the revised figure caption and main text.
>
> `Q3. How would the projection operator behave for complex or nonlinear constraints where analytical projectors are unavailable? Would this require solving an inner constrained least-squares problem at every step, and how does that impact computational cost?`
>
> A3.
>
> As explained in our general comment above and in our answer to W3, nonlinear or PDE-based constraints are handled by solving an inner least-squares projection at each step. This is illustrated in our data assimilation experiment for the Burgers equation of Section 5.2. This does not require exact convergence at every iteration: in practice, a small number of alternating steps is sufficient.  This design keeps the method scalable while preserving strict feasibility for the final sample.
>
>
> Q4. Given that the initialization step projects Gaussian noise to obtain (
> ), what guarantees exist that this projection produces feasible or representative samples, especially for high-dimensional or nonconvex manifolds?
>
> A4.
>
> Projecting Gaussian noise ensures feasibility by construction of the projection algorithm, and the guarantees therefore depend on the latter.
> While theoretical guarantees for nonconvex manifolds remain challenging, our empirical results on the Burgers PDE (with discontinuous manifolds) show that projected noise provides a valid initialization for the constrained sampling dynamics.
>
> Note however that feasibility and representativity of the first iterate $z_0$ is not central for our algorithm, and this first iterate is not expected to be close to the sampled physical signal. The output of our algorithm is the final sample $z_T$, which is the product of a sequence of coupled diffusion steps and projections. The first projection $z_0$ may be very far from physically plausible signals, and it is progressively brought closer to high-likelihood regions through the coupling with $x_t$.
>
>
> `Q5. How does your method perform in high-dimensional or multi-modal constrained settings? Does it exhibit the slow mixing behavior often observed in Langevin-based samplers, and what evidence supports stable convergence in such regimes?`
>
>
> A5.
> Our experiments demonstrate that SAL remains stable in high-dimensional multimodal settings.
>
> Our method is applied to sampling in high-dimensional and multimodel settings in the experiments, where the physical distributions are complex and multimodal. In particular, Section 5.1 investigates an experiment where the conditional distribution is multimodal by design.
>
> Note that although our theoretical framework is derived for Langevin Monte Carlo, our algorithm is applied to diffusion models for complex high-dimensional physical systems in our experiments. The implementation details are discussed in Appendix A.
>
> Therefore, SAL benefits the high-dimensional sampling abilities of diffusion models, which are evidenced in experiments 5.2 and 5.3. In those complex applications, we found sampling to be stable and efficient.
>
> A finite-time convergence analysis in Wasserstein space is provided in our answer to Reviewer 1KBT and complements these empirical findings.

---

### Author Response · Authors · 2025-11-15
**General Clarifications on Our Contributions**

Before addressing specific comments, we would like to highlight two overarching clarifications regarding the contributions of our work: **a conceptual comparison with other approaches**, and the **demonstration of SAL's ability to handle complex non-analytical projection operators**.

### Conceptual comparison with other approaches

To the best of our knowledge, SAL is the first method that enables diffusion models to be conditioned on non-convex hard constraints without introducing bias. This capability is important for many applications where strict physical constraints are essential.

Projection appears to be the necessary cost of obtaining exact constraint satisfaction Our analysis shows that soft penalties cannot enforce hard constraints, and that approaches relying on constraint satisfaction in expectation (e.g., Primal–Dual Langevin) also fail to ensure feasibility on individual samples.

By contrast, SAL maintains exploration while enforcing the constraint through a separate projected variable, coupled to the sample via a dual variable that converges (Proposition 5). This variable-splitting design gives SAL the ability to preserve the correct conditional distribution while enforcing constraint.

Below we provide a clearer mapping between the theoretical results (Propositions 3–6) and the empirical observations in Section 5, and we include a concise comparative summary table as recommended by reviewers gUbm and 1KBT.


#### (1) Strict constraint satisfaction

**Empirical observation**
Penalty and Primal–Dual Langevin methods exhibit constraint violations, as shown in Figure 2.

**Theoretical insight**
Proposition 3 shows that penalty methods optimize a Lagrangian with a fixed dual variable, leading to a suboptimal saddle point gap and hence infeasible samples [2].

**Our approahch**
SAL guarantees feasibility by explicitly projecting $z$ onto $\mathcal{C}$, achieving strict constraint satisfaction by design.

---

#### (2) Unbiased exploration

**Empirical observation**
Penalty methods and projected Langevin distort exploration due to repeated projections, as shown in Figures 2, 3 and 4.


**Our approach**
We decouple these two antagonistic objectives through variable splitting and coupling constraint.

**Theoretical insight**
The consistency between the two variables is ensured by imposing a coupling constraint. This constraint is provably achieved because the dual variable converges, as stated in Proposition 5.

---

#### (3) Comparative summary

| Algorithm | Constraint form | Strict constraint satisfaction | Unbiased exploration | Targets $p_\mathcal{C}$ | Duality convergence guarantee | Latent-space diffusion compatible |
|------------|----------------|-------------------------------|----------------------|-----------------------------|-----------------------------|-----------------------------------|
| Primal–Dual Langevin | Equality/inequality, on average | ✗ | ✓ | ✗ | ✓ | ✗ |
| Projected Langevin | Arbitrary sets | ✓ | ✗ | ✗ | ✗ | ✓ |
| Penalty / Guidance | Differentiable penalty | ✗ | ✗ | ✗ | ✗ | ✓ |
| **SAL** | Arbitrary sets | **✓** | **✓** | **✓** | **✓** | **✓** |


---

## General clarification on the complexity of projection operators

We thank the reviewers for pointing out that the constraint sets in our experiments might appear simple because some projection operators admit analytical forms. We acknowledge that this aspect was not sufficiently emphasized in the main text. In fact, Experiment 5.2 (data burgers equation) involves a non-analytical, high-dimensional, non-convex projection, implemented via an iterative numerical optimization procedure. Specifically, the feasible set is the intersection of mass and energy conservation constraints:

$$
\mathcal{C} = \{z \in \mathbb{R}^d \mid 1^{\top} z = M, \ \|z\|^2 = E^2 \},
$$

which defines a non-convex manifold. The projection step solves

$$
\underset{z \in \mathbb{R}^d}{\min} \ \frac{1}{2}\|z - x\|^2
\quad \text{subject to} \quad 1^{\top} z = M, \ \|z\|^2 = E^2,
$$

and is computed via an iterative alternating-projection algorithm, as described in Appendix E.2. This projection is significantly more complex than analytic projections, and is computed in dimension $d=200$ for a number of 50 iterations.

We also report in Appendix A.8 a quantitative study of runtime and computational cost, showing that the iterative projection introduces only a modest overhead relative to the diffusion score evaluation, which remains the dominant cost. Additionally, because strict feasibility is required only at the final iteration of SAL, intermediate projections can be computed approximately, providing a trade-off between computational cost and accuracy.

In the revised manuscript, we will move part of this discussion from the appendix to the main text (Section 5.2) to make this point clearer and to highlight that SAL naturally accommodates both analytical and numerical projection routines.

---

### Author Response · Authors · 2025-12-02
**Summary for the Area Chair**

We would like to provide the Area Chair with a concise summary of how the discussion unfolded and how the reviewers’ main concerns were addressed. Because the official reviews were reverted due to the system issue, this overview is intended to contextualize the reviewers’ initial impressions together with the substantial clarifications and additions made during the discussion.

---

### 1. Clarification of the main methodological concern

Several reviewers assumed that all constraint projections in our experiments were analytical. As clarified in the submission and reiterated during the discussion, **Experiment 5.2 already uses a non-analytical, high-dimensional, non-convex projection**, solved numerically through an iterative constrained optimization routine. This was present in the original manuscript, and we made it more explicit.

We also clarified our positioning relative to prior work, explaining how SAL differs from alternatives and why it is the only approach, to our knowledge, offering exact constraint satisfaction while preserving the exploration abilities of diffusion models.

During the discussion, we additionally provided:
- **finite-time convergence bounds** with hyperparameter guidance,
- **new large-scale PDE experiments** including Poisson, Darcy, and Navier–Stokes,
- **explicit quantitative tables** requested by multiple reviewers.

These additions directly addressed the core concerns raised across reviews.

---

## 2. Reviewers’ indications and our responses

### Reviewer zhci

**Main concerns:** breadth of related work, scope of experiments, and constraint complexity.

We clarified the relation to existing approaches and the relevance of the evaluation metrics used in physical systems. We emphasized that **Experiment 5.2 already features a complex, non-analytical projection** and provided further results on more challenging PDEs, showing the broad applicability of SAL.

---

### Reviewer 7Rvb

**Main concerns:** robustness to inexact projections, experimental breadth, hyperparameter ablations, and stability in latent space.

We noted that several of these elements were already present in the submission and expanded them with:
- sweeps over projection accuracy (residuals, bias proxies, runtime),
- concrete hyperparameter guidelines for both Langevin and latent diffusion sampling,
- a stability analysis identifying the latent-space instability threshold,
- new experiments on **Poisson**, **Darcy**, and **Navier–Stokes**, following the reviewer’s suggestions.

**Reviewer’s reaction during discussion:**
They updated their score from 2 to 4, expressed that the new theory and experiments addressed their concerns, and indicated that they would likely raise their score further after seeing the additional PDE experiments, which we then provided.

---

### Reviewer gUbm

**Main concerns:** limited breadth of experiments, comparison with baselines, quantitative evaluation, and runtime.

The reviewer noted that they would consider a higher rating if additional experiments were added. In response, we provided new experiments with explicit numerical tables (reconstruction error, constraint violation, runtime, projection accuracy), a clarified baseline comparison, and expanded runtime details.

---

### Reviewer 1KBT

**Main concerns:** need for experiments with complex projections and finite-time guarantees.

We clarified that Experiment 5.2 already uses a **numerically solved, high-dimensional, non-analytical projection** and provided a full **finite-time convergence analysis**, including:
- a Wasserstein proximal inequality,
- an $\mathcal{O}(\log T/\sqrt{T})$ convergence rate,
- stability conditions on step sizes.

We also added more PDE experiments with non-analytical constraints, further illustrating SAL’s applicability.

---

## 3. Current state of the paper after the discussion

The submission now includes:
- a clarified description of the non-convex projection in Experiment 5.2,
- new finite-time theoretical guarantees,
- new experiments on three additional high-dimensional PDE benchmarks, with complex constraints
- unified quantitative comparisons and metrics.

These additions substantially strengthen the work and directly respond to the reviewers’ feedback.

---

## 4. Context for the AC’s evaluation

Given the unusual situation with reverted reviews, we hope this summary helps situate the reviewers’ initial comments alongside the substantial clarifications, new theory, and new experiments provided during the discussion, as well as their stated intentions to update their assessments.

We appreciate the Area Chair’s consideration of the full exchange and thank you for evaluating the additional information.

---

### Meta-Review · Area_Chair_p4i6 · 2025-12-22

**Summary:**

This paper proposes Split Augmented Langevin (SAL), a training-free framework for sampling from generative models while strictly satisfying physical constraints. The method employs a variable-splitting strategy within a primal-dual framework, effectively decoupling the exploration of the data distribution from the strict enforcement of constraints via an augmented Lagrangian potential.

The reviewers consistently praised the theoretical rigor of the work, highlighting the principled variational formulation and the modular nature of the algorithm that allows it to work with pretrained diffusion models. The primary weakness identified across multiple reviews was the experimental breadth. The initial experiments were perceived as relying on low-dimensional problems with simple constraints, raising doubts about the method's utility for complex physical systems. This point is addressed to a large degree in rebuttal with new experiments for a number of more complicated, higher-dimensional systems. The AC recommendation is for a weak acceptance, noting that while the authors demonstrated the method's capability to handle complex constraints during the rebuttal, these extensive new results are currently not well integrated into the final manuscript and lacking details and explanations. This must be fixed if accepted.

**Reviewer Concerns:**

The concern regarding experimental breadth and constraint complexity was largely addressed by the author's inclusion of experiments on the Poisson equation, Darcy flow, and Navier-Stokes equations, along with quantitative tables comparing SAL to penalty-based and projected diffusion methods. The concern regarding the gap between measure-space theory and stochastic implementation, raised specifically by Reviewer 1KBT, was addressed through the provision of a new finite-time convergence analysis in the Wasserstein space. The concern regarding robustness to inexact projections was addressed by sensitivity studies showing performance stability under varying projection iterations. A remaining issue is the seamless integration of these new experiments into the main text, as well as lack of comparisons against broader landscape of constrained generation methods with authors excluding flow-based baselines.

**Reviewer Scores:**

Since the experiment breadth and metrics were universal asks by all reviewers and they were addressed by authors during rebuttal, their scores would likely all maintain or raise to the accepting range.

---

### Decision · Program_Chairs · 2026-01-26

Accept (Poster)